**DOI: 10.1038/ncomms14758**　**OPEN**

# Chd7 is indispensable for mammalian brain development through activation of a neuronal differentiation programme

Weijun Feng[1,*], Daisuke Kawauchi[2,*], Huiqin Körkel-Qu[1], Huan Deng[1], Elisabeth Serger[1], Laura Sieber[2], Jenna Ariel Lieberman[3], Silvia Jimeno-González[3], Sander Lambo[2], Bola S. Hanna[4], Yassin Harim[1], Malin Jansen[1], Anna Neuerburg[1], Olga Friesen[1], Marc Zuckermann[4], Vijayanad Rajendran[2], Jan Gronych[4], Olivier Ayrault[5], Andrey Korshunov[6,7], David T.W. Jones[2,7], Marcel Kool[2], Paul A. Northcott[8], Peter Lichter[4,7], Felipe Cortés-Ledesma[3], Stefan M. Pfister[2,7,9] & Hai-Kun Liu[1]

Mutations in chromatin modifier genes are frequently associated with neurodevelopmental diseases. We herein demonstrate that the chromodomain helicase DNA-binding protein 7 (Chd7), frequently associated with CHARGE syndrome, is indispensable for normal cerebellar development. Genetic inactivation of *Chd7* in cerebellar granule neuron progenitors leads to cerebellar hypoplasia in mice, due to the impairment of granule neuron differentiation, induction of apoptosis and abnormal localization of Purkinje cells, which closely recapitulates known clinical features in the cerebella of CHARGE patients. Combinatory molecular analyses reveal that Chd7 is required for the maintenance of open chromatin and thus activation of genes essential for granule neuron differentiation. We further demonstrate that both Chd7 and Top2b are necessary for the transcription of a set of long neuronal genes in cerebellar granule neurons. Altogether, our comprehensive analyses reveal a mechanism with chromatin remodellers governing brain development via controlling a core transcriptional programme for cell-specific differentiation.

[1] Division of Molecular Neurogenetics, German Cancer Research Center (DKFZ), DKFZ–ZMBH Alliance, Im Neuenheimer Feld 280, Heidelberg 69120, Germany. [2] Division of Pediatric Neuro-oncology, German Cancer Research Center (DKFZ), Im Neuenheimer Feld 280, Heidelberg 69120, Germany. [3] Centro Andaluz de Biología Molecular y Medicina Regenerativa (CABIMER), CSIC-Universidad de Sevilla-Universidad Pablo de Olavide, Sevilla 41092, Spain. [4] Molecular Genetics, German Cancer Research Center (DKFZ), Im Neuenheimer Feld 280, Heidelberg 69120, Germany. [5] Institut Curie, CNRS UMR 3347, INSERM U1021, Centre Universitaire, Bâtiment 110, 91405 Orsay, France. [6] Clinical Cooperation Unit Neuropathology, German Cancer Research Centre (DKFZ), Department of Neuropathology, University of Heidelberg, Heidelberg 69120, Germany. [7] German Cancer Consortium (DKTK), Core Center Heidelberg, Heidelberg 69120, Germany. [8] Department of Developmental Neurobiology, St. Jude Children's Research Hospital, 262 Danny Thomas Place, Memphis, Tennessee 38105, USA. [9] Department of Pediatric Hematology and Oncology, Heidelberg University Hospital, Im Neuenheimer Feld 430, 69120 Heidelberg, Germany. * These authors contributed equally to this work. Correspondence and requests for materials should be addressed to D.K. (email: d.kawauchi@dkfz.de) or to S.M.P. (email: s.pfister@dkfz.de) or to H.-K.L. (email: l.haikun@dkfz.de).

The chromatin structure plays a central role in any DNA-dependent biological process such as transcription, DNA replication and DNA repair. Recurrent mutations of chromatin modifier genes have been identified by extensive high-throughput sequencing of a broad spectrum of human patient samples. These findings reveal perturbation of the epigenome as a new hallmark of cancer[1] and neurological diseases[2]. This strong correlative evidence requires further functional validation to fully understand the role of chromatin modifiers in development and disease, however.

Chromatin remodellers are key players in the regulation of nucleosome positioning and thus control DNA accessibility in eukaryotic cells. Mutations in the ATP-dependent chromatin remodeller chromodomain, helicase, DNA binding (CHD) 7 are the major cause of CHARGE syndrome, which is characterized by multiple organ defects including Coloboma, Heart defect, Atresia choanae, Retarded growth and development, Genital abnormality and Ear abnormality[3]. Notably, many CHARGE syndrome patients have brain anomalies including hypoplasia of the olfactory bulb and cerebellum[4,5] and often display an autistic phenotype[6]. Moreover, CHD7 mutations have been identified in patients with Kallmann syndrome[7], autism spectrum disorders (ASD)[8] as well as subsets of patients with both congenital heart disease and neurodevelopmental disabilities[9]. These findings strongly suggest an important role of CHD7 in brain development. Heterozygous Chd7 gene trap mice have exhibited many CHARGE-relevant phenotypes[4,10,11], supporting the role of transgenic mice as a disease-relevant tool. Our studies and others have utilized Chd7 conditional knockout mice to specifically dissect the role of Chd7 in the brain, demonstrating that mutations of Chd7 could be a causal factor for olfactory defects as well as hearing loss and cognitive disabilities in CHARGE patients[12–14]. However, the exact role of Chd7 in cerebellar development remains unclear.

In both human and mice, the cerebellum undergoes a rapid expansion and foliation during early postnatal age. This expansion of the cerebellum is primarily due to the fast proliferation of cerebellar granule neuron progenitors (GNPs) in the outer external granule layer (oEGL), which is stimulated by Sonic hedgehog (Shh) secreted from Purkinje cells[15]. On the other hand, GNPs secret a glycoprotein Reelin (Reln) to control the proper localization of Purkinje cells[15]. In a precisely and temporally controlled fashion, GNPs shift to the inner EGL (iEGL) coincident with exiting the cell cycle[15]. The postmitotic cerebellar granule neurons (thereafter called as CGNs) then migrate inwardly to reach the internal granule layer (IGL), where they become mature. Importantly, unbalance between GNP proliferation and differentiation can lead to cerebellar hypoplasia or tumour formation[16]. Emerging data demonstrate that chromatin landscape is dynamically changed during differentiation of cerebellar granule cells, implicating an important role of chromatin regulation in neuronal differentiation[17,18]. Indeed, cerebellar anomalies including vermis hypoplasia and massive Purkinje cell heterotopia represent some of the most prominent features of CHARGE patients[4,19], strongly suggesting that CHD7 is essential for normal cerebellar development.

Here, we report in vivo evidence for an essential role of Chd7 during cerebellar development. Using genetic and biochemical approaches, we unravel molecular mechanisms of how Chd7 controls neuron differentiation to govern proper formation of the cerebellar architecture. These results provide not only new molecular insights into cellular differentiation, but also a better understanding of Chd7-associated human diseases.

## Results

**Chd7 is highly expressed in cerebellar granule cells.** Many CHARGE patients carrying CHD7 mutations have defects in the cerebellum, implicating a functional role of CHD7 in cerebellar development. In support, expression of CHD7 is enriched in the cerebellum during human brain development when compared to the other brain regions (Supplementary Fig. 1a). To investigate the contribution of Chd7 to cerebellar development, we examined detailed spatial and temporal expression patterns of Chd7 in developing mouse cerebellum. Immunohistochemistry (IHC) with a Chd7 antibody at embryonic day (E) 14.5 revealed a higher expression of Chd7 in the external granule layer (EGL) as compared to cells in the ventricular zone (Fig. 1a). Chd7 is highly expressed in cells from EGL and IGL at neonates and persists in the IGL at P21 cerebella (Fig. 1a). A very similar expression pattern of GFP was observed in a bacterial artificial chromosome-based Chd7-GFP reporter mouse line carrying GFP fused to the Chd7 regulatory element (Fig. 1a, lower panels). Consistent with the expression of Chd7 protein, in situ hybridization data revealed that Chd7 mRNA was clearly detected in EGL at E14.5, and in the granule layer at P7 and adult cerebellum (Supplementary Fig. 1b). We next performed co-immunostaining assays to check the co-expression of Chd7 or GFP with cell type-specific markers in the cerebella of wild-type (WT) or Chd7-GFP reporter mice, respectively. Exampled as P7 cerebellum, over 90% of Pax6[+] or Pax2[+] cells, namely cerebellar granule cells or inhibitory interneuron progenitors, respectively, expressed GFP in Chd7-GFP mice (Fig. 1b). In contrast, Bergmann glia (Sox2[+]) and Purkinje cells (Calbindin[+]) only showed a very weak expression of Chd7 (Fig. 1c).

Intriguingly, higher levels of Chd7 were detected in iEGL, where cells have just exited cell cycle (p27[+]) as compared to proliferating GNPs (Ki67[+]) in oEGL (Fig. 1d). In support, quantitative RT–PCR (qRT-PCR) data of cultured granule cells validated a higher expression of Chd7 in postmitotic (Ccnd1-low) cells than proliferating (Ccnd1-high) cells (Fig. 1e). These data suggest that the expression of Chd7 is upregulated during the transition from proliferating into differentiated cerebellar granule cells.

**Disruption of Chd7 in GNP leads to cerebellar hypoplasia.** To examine roles of Chd7 during cerebellar development, we next genetically ablated Chd7 in vivo using Chd7 conditional knockout mice[12]. We first utilized a Nestin-Cre driver to target neural stem cells in the brain. The homozygous mutant [Nestin-Cre::Chd7f/f] mice exhibited largely perinatal lethality (Supplementary Fig. 2a). We observed conspicuous cerebellar hypoplasia in survived homozygous mutant mice (Supplementary Fig. 2b). To dissect the role of Chd7 in cerebellar development and overcome the lethality of Nestin-Cre-mediated Chd7 deletion, we decided to use cerebellar neuron-specific Cre drivers. The cerebellar neuroepithelium (Nestin[+]) is mainly divided into two compartments during embryogenesis: the upper rhombic lip (Atoh1[+]) and the cerebellar ventricular zone (Ptf1a[+]) that give rise to excitatory (for example, granule cells) and inhibitory neurons (for example, Purkinje cells), respectively[15]. Given that Chd7 is predominantly expressed in granule cells (Fig. 1), we first ablated Chd7 in Atoh1[+] cells. IHC assays with a Chd7 antibody performed in P7 cerebella (Supplementary Fig. 3a) and in vitro cultured granule cells (Supplementary Fig. 3b) verified ablation of Chd7 in granule cells of [Atoh1-Cre::Chd7f/f] homozygous mutant mice. While no structural anomalies of the cerebella were observed at E15.5, P0 and older [Atoh1-Cre::Chd7f/f] homozygous mutant mice, but not [Atoh1-Cre::Chd7f/+] heterozygous mutant cerebella showed severe defects in folia formation and cerebellar hypoplasia especially in the vermis (Fig. 2a, Supplementary Fig. 3c). The cerebellar defects in Chd7 mutant animals are more prominent in the anterior lobe (Fig. 2a). This is consistent with the previous

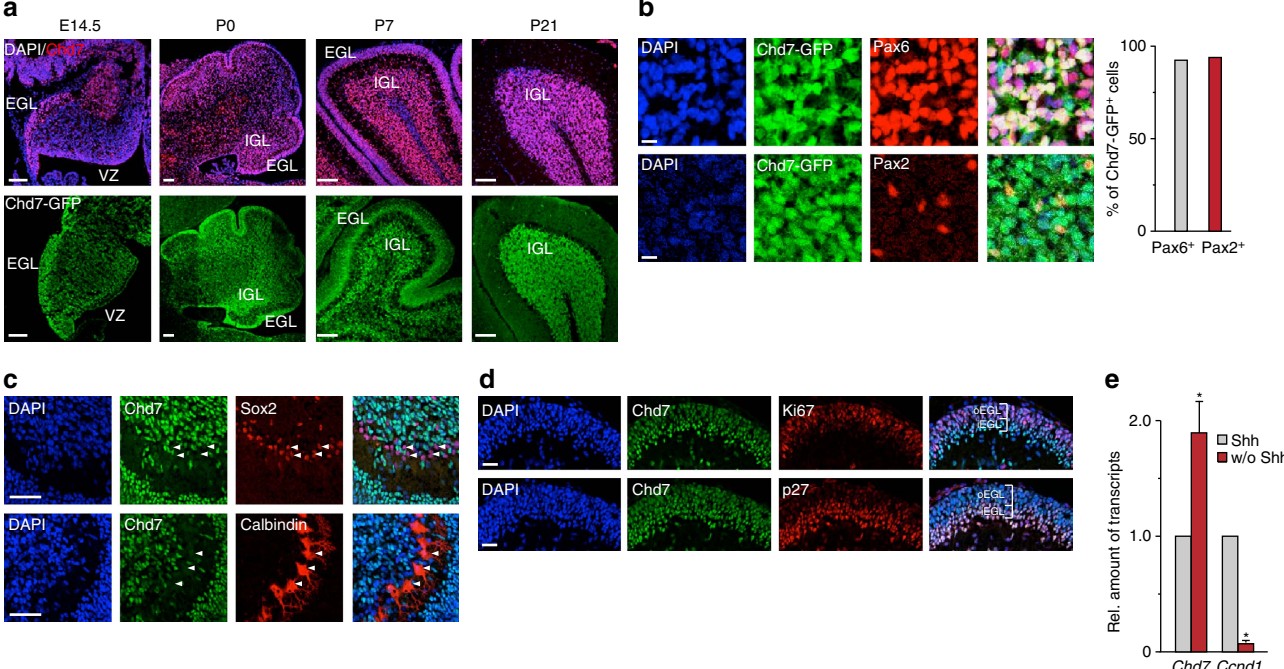

**Figure 1 | Chd7 is highly expressed in cerebellar granule cells.** (**a**) Immunostaining of Chd7 (red, upper panels) and Chd7-GFP (green, lower panels) in a time series of mouse cerebellum during development. Cellular nuclei were counterstained with DAPI (blue). Scale bars, 100 μm. EGL, external granule layer; IGL, internal granule layer. (**b**) Co-immunostaining of GFP (green) with Pax6 (red, upper panels) and Pax2 (red, lower panels) in the IGL of cerebella of P7 Chd7-GFP reporter mice. Scale bars, 10 μm. The bar graph shows the quantification of percentages of Chd7-GFP+ cells either in Pax6+ or Pax2+ cells. (**c**) Co-immunostaining of Chd7 with Sox2 (red, upper panels) and Calbindin (red, lower panels) in cerebella of P7 wild-type C57BL/6J mouse. Arrowheads show very weak staining of Chd7 in Sox2+ or Calbindin+ cells. Scale bars, 50 μm. (**d**) Co-immunostaining of Chd7 (green) with Ki67 (red, upper panels) and p27 (red, lower panels) in EGL of P7 cerebellum. Scale bars, 25 μm. oEGL: outer EGL. iEGL: inner EGL. (**e**) Quantitative reverse transcription PCR (qRT–PCR) results show the expression level of Chd7 and Ccnd1 in granule neuron progenitors (GNPs) cultured with or without (w/o) Sonic Hedgehog (Shh) for 48 h. The level of Gapdh transcripts was used for normalization. Bars represent normalized mean value ± s.d. from three independent samples. Paired two-tailed t-test with equal variance was performed, $P = 0.031$ (for Chd7), $P = 0.0001$ (for Ccnd1).

report[20] and a remaining Chd7 protein in the posterior lobe (Supplementary Fig. 3d), which is probably due to the less recombination efficiency of the *Atoh1-Cre* mouse line.

The crosstalk between GNPs and Purkinje cells is essential for cerebellar development, as revealed by severe cerebellum hypoplasia in *Reln*-deficient *reeler* mice[21]. Since massive Purkinje cell heterotopia has been reported in CHARGE patient brains[19,22], we next examined Purkinje cells in [*Atoh1-Cre::Chd7^{f/f}*] mutant mice. Although the *Atoh1-Cre* driver does not target Purkinje cells, abnormal distribution of Purkinje cells were clearly detected in cerebella from P0 onwards until adult, especially at the anterior lobe of the cerebellum (Fig. 2b, arrowheads). Importantly, we did not observe any abnormal phenotype of Purkinje cells, when directly targeting *Chd7* in Purkinje cells using a *Ptf1a-Cre* driver[23] (Fig. 2c). Therefore, our findings suggest that the Purkinje cell phenotype in *Chd7* mutant mice and very likely in CHARGE patients is not due to a Purkinje cell-autonomous effect, but rather results from *Chd7* loss in granule cells.

**Chd7 regulates neuronal differentiation and cell survival.** To dissect cellular mechanisms responsible for the cerebellar hypoplasia phenotype of *Chd7*-deficient mice, we analysed cell proliferation, differentiation, cell death and migration of granule cells during cerebellar development. Two-hour *in vivo* pulse-labelling of BrdU or IdU revealed similar percentages of proliferating GNPs between *Chd7* WT [*Chd7^{f/f}*] and *Chd7*-deficient [*Atoh1-Cre::Chd7^{f/f}*] cerebella at both P0 and

P3 (Fig. 3a). As further validation, we freshly isolated GNPs at P7 using a percoll gradient centrifugation[24] after pulse-labelling of BrdU 2 h before the killing. The FAC-based cell cycle analysis revealed no significant change of the ratio of proliferating cells between WT and *Chd7*-deficient mice (Supplementary Fig. 3e). Together, these data revealed that the proliferation capacity of *Chd7*-deficient GNPs seems comparable to WT cells. However, 24-h BrdU labelling at P6 followed by co-staining of BrdU and Ki67 at P7 showed less Ki67-negative cells among BrdU-positive cells in the EGL of *Chd7*-deficient mice, indicating that *Chd7* loss prevents appropriate cell cycle exit of GNPs *in vivo* (Fig. 3b). Failure of cell cycle exit of *Chd7* mutant GNPs implicates less differentiated granule cells. Indeed, IHC analysis with a neuronal marker NeuN revealed significantly less granule neurons in the IGL of *Chd7* mutant than WT cerebella (Fig. 3c). The decrease of granule neurons in *Chd7* mutant cerebella was confirmed by the counting of BrdU+ cells in IGLs of P7 pups that were pulsed and traced with BrdU for 48 h (Supplementary Fig. 3f). Moreover, IHC for cleaved caspase3 detected more apoptotic cells in the *Chd7*-deficient EGL at postnatal stages (Fig. 3d). In line with this observation, *Chd7* loss significantly increased cell death as detected by staining of cleaved caspase3 and Parp in cultured granule cells especially in absence of SHH (Fig. 3e), indicating that mutant cells are more prone to cell death upon differentiation. Meanwhile, we did not observe defects in the migration of granule cells in *Chd7*-deficient cerebella, as shown by BrdU labelling and tracing (for 48 h) assay (Supplementary Fig. 3f). Thus, our data suggest that cerebellar hypoplasia in [*Atoh1-Cre::Chd7^{f/f}*] mutant mice results from both granule

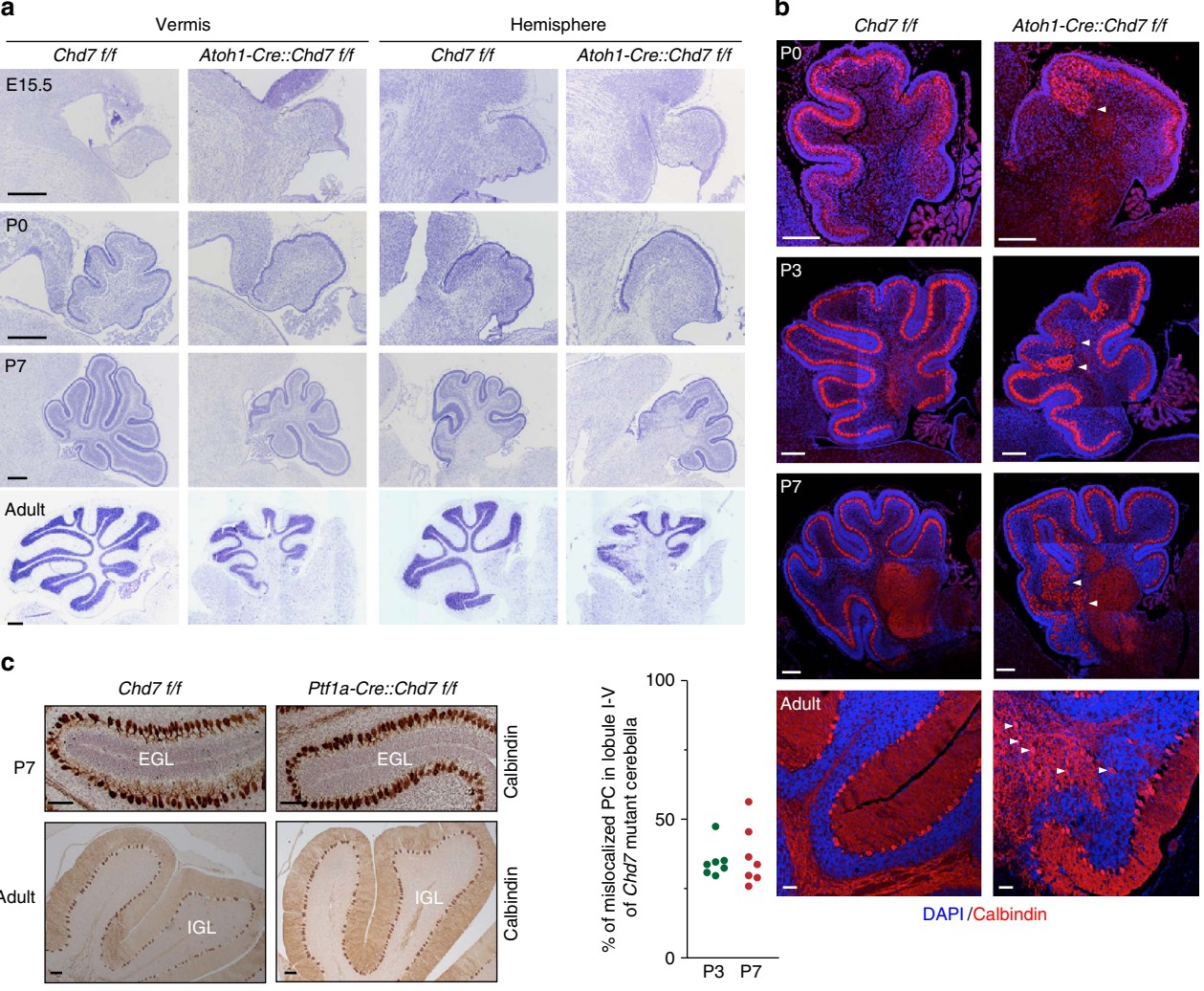

**Figure 2 | Genetic ablation of *Chd7* in GNPs leads to cerebellar hypoplasia and Purkinje cell ectopia. (a)** Nissl staining of E15.5, P0, P7 and adult cerebella including vermis and hemisphere from [*Chd7f/f*] and [*Atoh1-Cre::Chd7f/f*] mice. Left side in each panel is the anterior part of cerebella. Scale bars, 100 μm. **(b)** Co-immunostaining of Calbindin (red), a marker for Purkinje cells (PCs) in P0, P3, P7 and adult WT [*Chd7f/f*] and *Chd7* mutant [*Atoh1-Cre::Chd7f/f*] cerebella. Cellular nuclei are counterstained with DAPI (blue). Arrowheads indicate clusters of mislocalized PCs. Scale bars, 200 μm. Quantification of the percentage of mislocalized PCs among all PCs in anterior lobule I-V is shown in the left panel. Cerebellar sections from three independent P3 and six P7 *Chd7* mutant mice were counted. **(c)** Immunostaining of Calbindin in cerebella of P7 and adult WT [*Chd7f/f*] and *Chd7* mutant [*Ptf1a-Cre::Chd7f/f*] mice. Sections were counterstained with haematoxylin. Scale bars, 100 μm.

cell-autonomous effect (that is, a failure of GNP differentiation and increased cell death) and non-autonomous effects from the abnormal distribution of Purkinje cells.

***Chd7* loss alters expression of neuronal genes**. To explore molecular consequences of *Chd7* loss in GNPs, we next performed RNA sequencing (RNA-seq)-based transcriptome analysis using freshly isolated GNPs from P7 *Chd7* WT, heterozygous [*Atoh1-Cre::Chd7f/+*] (*Chd7* Het.) and homozygous [*Atoh1-Cre::Chd7f/f*] (*Chd7* Hom.) mutant cerebella. Results from qRT–PCR showed a significant decrease of *Chd7* transcripts according to their genotypes in *Chd7* Het. and *Chd7* Hom. mutant GNPs as compared to WT GNPs (Fig. 4a). At the significance of $P < 0.01$, 151 genes were downregulated in *Chd7* Hom. GNPs, with many of them (62.9%, 95/151, $P < 0.05$ between WT and *Chd7* Het.) significantly downregulated in *Chd7* Het. GNPs (Fig. 4b), suggesting a dosage-dependent regulation of these genes by Chd7. Importantly, DAVID Gene

Ontology (GO) term analysis of genes downregulated upon loss of *Chd7* demonstrated neuronal functions including synaptic transmission, ion transport, neuron development and neuron differentiation as the most enriched biological processes (Fig. 4c), while no significantly enriched term (Benjamini adjusted $P$ value $< 0.05$) was identified in genes significantly upregulated upon *Chd7* loss. For instance, the proper level of Chd7 is necessary for the expression of neural genes that are important for cerebellar development, including *Cadps2*, *Gap43*, *Neurod1*, *Reln* and *Unc5c* (refs 25–29) (Fig. 4d, Supplementary Fig. 4a). Consistent with downregulation of Reln, IHC analyses demonstrated that Dab1, the adaptor protein of Reln signalling pathway, was upregulated specifically in mislocalized Purkinje cells in *Chd7* mutant cerebella (Supplementary Fig. 4b) in accordance with observation in the *reeler* mice[21]. To exclude the long-term or indirect effect of Chd7 loss on gene expression, we acutely ablated the expression of Chd7 by treating [*Chd7f/f*] or WT granule cells with cell-permeable Cre for 48 h. Results from qRT–PCR showed a consistent downregulation

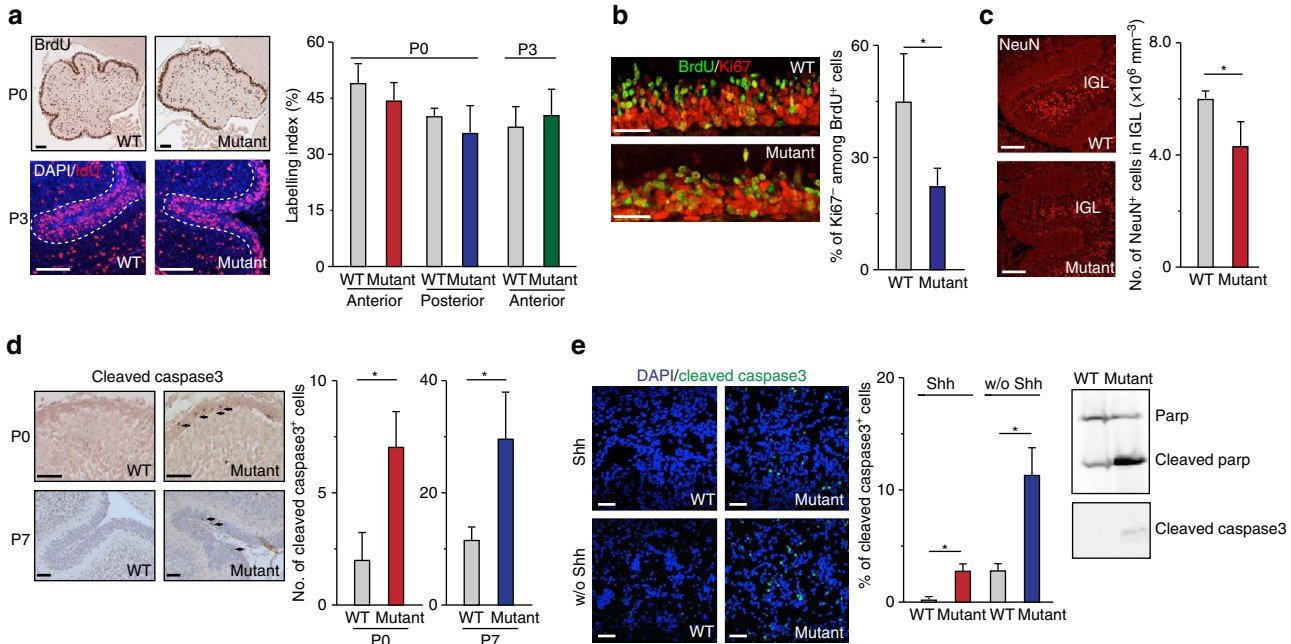

**Figure 3 | Mutation of *Chd7* in GNPs impairs neuronal differentiation and leads to cell death. (a)** Immunostaining of BrdU (brown) in P0 and IdU (red) in P3 cerebella from WT [*Chd7^f/f*] and *Chd7* mutant [*Atoh1-Cre::Chd7^f/f*] mice pulse-labelled with BrdU or IdU for 2 h before the killing. The dashed lines show the inner border of EGL in P3 cerebella. Scale bars, 100 μm. Quantification of the percentage of BrdU- or IdU-positive cells in the EGL within selected areas is shown in the right panel. Bars represent mean value ± s.d. from four mice for each group (for P0) and three mice for each group (for P3). **(b)** Co-immunostaining of BrdU (green) and Ki67 (red) in anterior cerebellar lobules of P7 WT and *Chd7* mutant mice subjected to BrdU administration 24 h before the killing. Scale bars, 50 μm. The graph represents the percentage of Ki67-negative cells among BrdU-positive cells. Bars represent mean value ± s.d. from three mice for each group. $P = 0.047$. **(c)** Immunostaining of NeuN in anterior cerebellar lobules of P3 WT and *Chd7* mutant mice. Scale bars, 50 μm. Quantification of the number of NeuN⁺ cells in IGL is shown in the right panel. Bars represent mean value ± s.d. from three mice for each group. $P = 0.035$. **(d)** IHC of cleaved-caspase3 in P0 and P7 cerebella of WT and *Chd7* mutant mice. Scale bars, 50 μm. The number of cleaved caspase3-positive cells (indicated by arrows) in the whole EGL of cerebellar sections was counted (right panels). Bars represent mean value ± s.d. from four mice for each group (for P0, $P = 0.0027$) and three mice for each group (for P7, $P = 0.024$). **(e)** Immunostaining of cleaved caspase3 in WT and *Chd7* mutant GNPs cultured with or without (w/o) Shh for 48 h (left panels). The percentage of cleaved caspase3-positive cells (green) among DAPI-stained cells (blue) was quantified (middle panel). Scale bars, 50 μm. The expression level of Parp, cleaved parp and cleaved caspase3 in WT and *Chd7* mutant GNPs cultured without Shh shown by western blots (right panels). Bars represent mean value ± s.d. from five randomly selected areas. $P = 2.1E-5$ (for Shh); $P = 7.7E-5$ (for w/o Shh).

of Chd7 regulated genes identified above (Supplementary Fig. 4c).

To identify direct target genes of Chd7, we performed chromatin-immunoprecipitation coupled to sequencing (ChIP-seq) using freshly isolated P7 GNPs. In order to identify *bona fide* Chd7-binding sites, we in parallel performed Chd7 ChIP-seq on *Chd7*-deficient GNPs. Using the MACS (*M*odel-based *A*nalysis of *C*hIP-seq) peak calling method, 1,946 peaks were called from WT cells, in contrast to only 96 peaks called from *Chd7*-deficient cells. In consistency with reports showing that Chd7 preferentially binds to enhancers in ESCs and oligodendrocytes[30,31], Chd7-occupied regions in GNPs are enriched of H3K27ac, a histone mark for active promoters and enhancers (Fig. 4e). Furthermore, Chd7 was enriched in super-enhancers (SEs) defined by H3K27ac ChIP-seq on GNPs (ref. 31) (Fig. 4f,g, Supplementary Fig. 5b). GO analysis revealed a strong enrichment of *Chd7*-associated genes in biological processes like transcription, chromatin organization, neuron differentiation and neuron projection morphogenesis (Supplementary Fig. 5a). Among genes whose expression was significantly altered upon loss of *Chd7* ($P < 0.05$), 12.5% (93/746) downregulated and 9.9% (99/1002) upregulated genes showed association of Chd7. For instance, Chd7 binds Chd7-activated genes, including *Neurod1*, *Cadps2*, *Gap43* and *Reln* (Fig. 4g, Supplementary Fig. 5b).

As Chd7 functions as an ATP-dependent nucleosome remodeller[32], we hypothesized that Chd7 regulates transcription via controlling DNA accessibility of *cis*-regulatory elements. For this, we utilized an assay for Tn5 transposase-accessible chromatin with high-throughput sequencing (ATAC-seq) (ref. 33) and generated a genome-wide map of the open chromatin landscape in *Chd7* WT and mutant GNPs at P7. As expected, most of H3K27ac ChIP-seq peaks (81.9%) colocalized with ATAC-seq peaks (Supplementary Fig. 6a). A small subset of ATAC-seq peaks showed significant alteration upon loss of *Chd7* (2.1%), with *Chd7* Het. cells exhibiting intermediate change between WT and *Chd7* Hom. (Supplementary Fig. 6b). GO analysis of genes containing altered ATAC-seq peaks called transmission of nerve impulse, cell motion and cell adhesion as the most enriched biological processes (Supplementary Fig. 6c). Among genes whose expression was significantly changed upon loss of *Chd7* ($P < 0.05$), 14.2% (106/746) downregulated and 4.5% (45/1002) upregulated genes showed altered ATAC-seq peak. For instance, decreased ATAC-peaks upon *Chd7* loss were detected in Chd7-activated genes including *Cadps2*, *Nkain3*, *Reln*, *Bmp5* and *Fstl5* (Fig. 4h, Supplementary Fig. 6d). Moreover, 22.8% (320/1402) of genes bound by Chd7 in GNPs showed altered ATAC-seq peaks upon *Chd7* loss, revealing that the association of Chd7 is required for the maintenance open chromatin on these genes. Collectively, comprehensive analysis of ChIP-seq and ATAC-seq data (for example, Supplementary Table 1) demonstrate that *Chd7* is involved in the maintenance of open chromatin at the regulatory elements of genes related with neuronal differentiation.

**Chd7 regulates granule cell differentiation with Top2b**. To get insight into molecular mechanisms underlying Chd7-dependent transcriptional regulation, we aimed at identifying CHD7-interacting proteins using immunoprecipitation (IP) coupled with mass spectrometry (MS). For efficient IP with a sufficient amount

of exogenous CHD7 protein, a Flag-tagged, truncated CHD7 (1–1,899 amino acids) that preserves nucleosome remodelling activity *in vitro*[32] was overexpressed and immunoprecipitated in *CHD7*-deficient HEK293T cells (Supplementary Fig. 7a,b). Besides previously published CHD7-interacting partners, such as

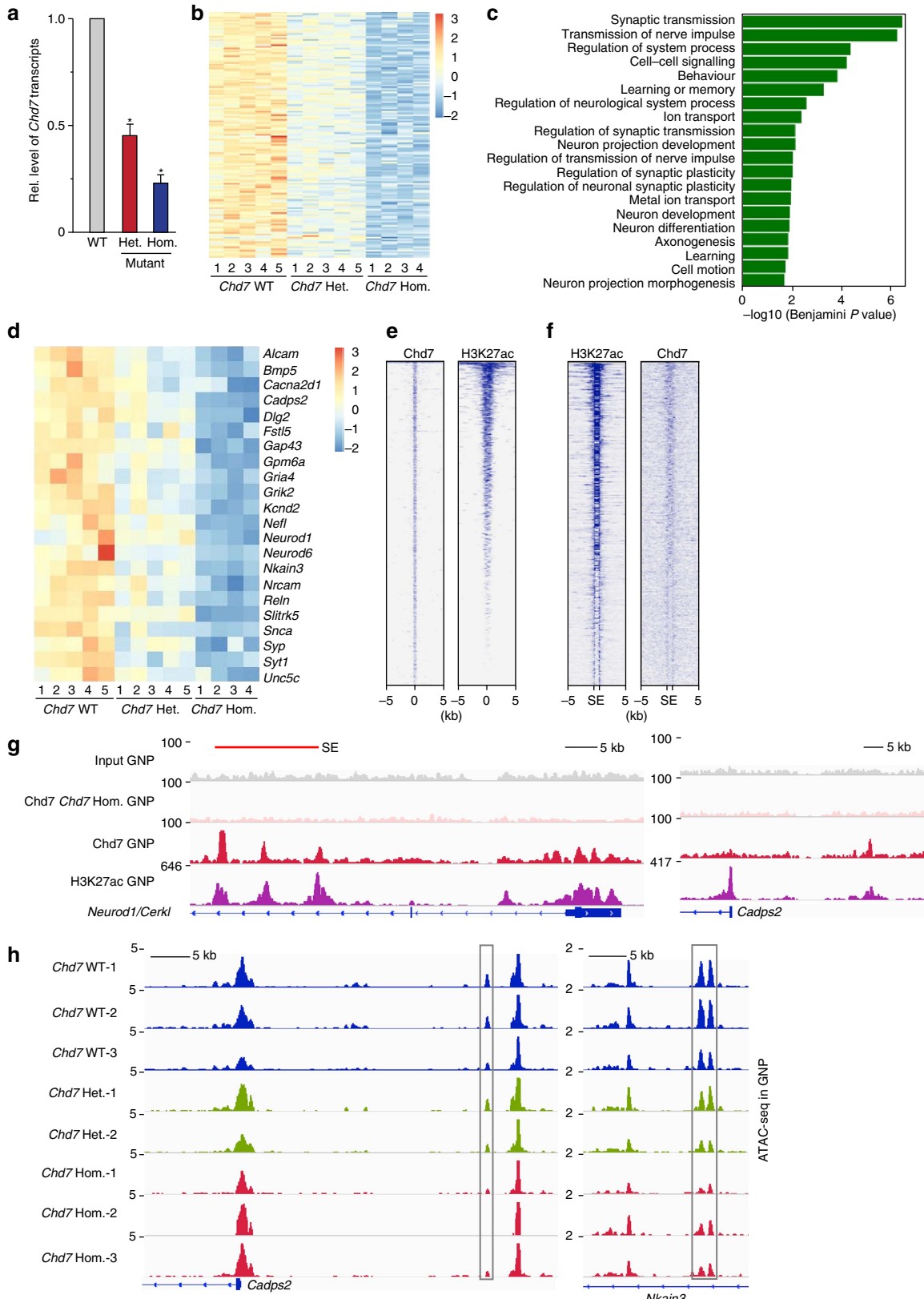

BRG1, PARP1 and CHD8 (refs 34,35), DNA topoisomerases TOP2A, TOP2B and TOP1 were identified by MS (Fig. 5a). Protein–protein interactions between CHD7 with these proteins were validated by co-IP assays using nucleic acids-free nuclear extracts (Supplementary Fig. 7c) from HEK293T cells expressing full-length Flag-tagged CHD7 (Fig. 5b). DNA topoisomerases are key enzymes that solve the topological problems occurring during virtually all DNA-related processes[36]. Inhibition of enzymatic activities of DNA topoisomerases in postmitotic neurons selectively represses the expression of long genes[37,38]. Top2b was shown to be important for neuronal migration in the developing neocortex by controlling the expression of Reln (ref. 39). Therefore, we decided to further explore the functional relevance of interaction between Chd7 and Top2b. Co-IP assays showed the interaction between endogenous Top2b and Chd7 proteins in not only HEK293T cells but also adult cerebellar cells, 85–90% of which consist of CGNs (ref. 40) (Fig. 5c). We also demonstrate that the N-terminal part (1–786 aa) of CHD7 is responsible for its interaction with TOP2B, whereas KU70 does not interact with the N-terminus of CHD7 and PARP interacts with several regions of CHD7 (Fig. 5d).

Previous immunostaining analysis demonstrated that GNPs in the oEGL express Top2a, whereas differentiating cells in iEGL and differentiated CGNs in IGL only express Top2b (ref. 41). Consistently, the transcript level of Top2a in cultured granule cells drastically decreased upon differentiation of proliferating GNPs (cultured with Smoothened agonist SAG) into postmitotic (cultured without SAG) CGNs, as analysed using Affymetrix microarray (Supplementary Fig. 8a). As Top2b is the sole DNA topoisomerase II expressed in CGNs, we decided to use in vitro induced postmitotic CGN to dissect the functional interaction between Chd7 and Top2b. We performed immunostaining of granule cells at 1 and 48 h after plating in order to compare the percentage of proliferating and differentiated cells between WT and Chd7-deficient cells. At 1 h after plating, over 80% of cells were proliferating (Ki67$^+$) and expressed Pax6 (Supplementary Fig. 8b). In contrast, after 48 h, most of cells exited cell cycle (Ki67$^-$) and expressed p27, suggesting that these cells were newly differentiated in culture (Supplementary Fig. 8c). Importantly, no statistical difference of percentages of proliferating and differentiated cells between WT and Chd7 mutant cells at these two time points were observed (Supplementary Fig. 8b,c). To identify genes regulated by Top2b in CGNs, we performed microarray-based gene expression analysis using cultured CGNs treated with DMSO as a control or ICRF-193 (10 μM for 12 h), an enzymatic inhibitor of Topoisomerase II (ref. 42). Consistent with previous reports in cortical neurons[37,38], long genes (gene length > 100 kb) were selectively downregulated in CGNs upon treatment of ICRF-193 (Fig. 6a, left panel). The average size of genes downregulated by ICRF-193 treatment ($P < 0.05$, fold change > 1.5) is approximately 300 kb, while the average sizes of all genes expressed in CGNs and the genes upregulated upon ICRF-193 treatment are both around 50 kb (Supplementary

Fig. 9a). To directly compare with Top2b-dependent transcription, we next identified Chd7-regulated genes by microarray analysis using Chd7 WT and homozygous mutant CGNs. Interestingly, many long genes were significantly downregulated upon loss of Chd7 (Fig. 6a, right panel). We subsequently performed GSEA (Gene set enrichment analysis) to check the concordant regulation of gene expression by Top2b and Chd7. Genes that were significantly downregulated upon ICRF-193 treatment showed lower expression level in Chd7-deficient CGNs than WT cells (Fig. 6b, left panel). Also, genes that were significantly downregulated in Chd7-deficient CGNs were largely repressed upon ICRF-193 treatment (Fig. 6b, right panel). At the significance of $P < 0.05$, 240 genes were commonly activated by Chd7 and Top2b in CGNs (Supplementary Fig. 9b). Intriguingly, the average gene length of Top2b and Chd7 co-activated genes was even longer than those genes activated by Top2b but not Chd7 (Supplementary Fig. 9b), implicating that transcription of these very long genes requires both enzymes. Next, we compared expression levels of these 240 genes between GNPs cultured with and without SAG. Most of these 240 genes (84.6%, 203/240) were significantly upregulated upon differentiation (Fig. 6c). Using qRT–PCR, we validated the downregulation of ASD-related genes involved in cerebellar development (Cadps2 and Reln) (refs 28,43), and ASD- or epilepsy-related ion channels in neurons (Cacna2d1, Grik2 and Kcnd2) (refs 43–45) in ICRF-193 treated or Chd7 mutant CGNs (Fig. 6d,e). Of note, Chd7 expression in CGNs was significantly downregulated upon ICRF-193 treatment (Fig. 6d), indicating that Top2b is required for activation of Chd7 during differentiation.

We hypothesized that Chd7 may be required for proper targeting of Top2b to chromatin. To test this, we performed Top2b ChIP-seq in WT and Chd7-deficient CGNs. The expression level of Top2b in CGNs was not altered upon loss of Chd7, at the level of both transcript and protein (Supplementary Fig. 9c). The specificity of the Top2b antibody was verified with ChIP analysis on WT and Top2b$^{-/-}$ mouse embryonic fibroblasts[46] using primer sets for Top2b-targeted regions at the Fos gene (Supplementary Fig. 9d)[47]. We next aimed at comparing the genome-wide binding patterns of Top2b and Chd7. Since Chd7 ChIP-seq was done on GNPs, we performed ChIP-qPCR assays to validate the binding of Chd7 in CGNs. Using primers amplifying Chd7 binding regions identified by Chd7 ChIP-seq in GNPs, we detected the association of Chd7 with multiple regions in WT CGNs, and a clear decrease of its association in Chd7 mutant CGNs (Supplementary Fig. 9e). The comparison between the binding pattern of Chd7 in GNPs and that of Top2b in CGNs revealed that Top2b is associated with Chd7 binding sites (Fig.6f, Supplementary Fig. 9f). While the average length of genes bound by Chd7 was around 103 kb, the average gene length of genes co-occupied by Chd7 and Top2b was approximately 165 kb, revealing preferential binding of Chd7 and Top2b on long genes in cerebellar granule cells. Importantly, upon loss of Chd7,

**Figure 4 | Chd7 is required for activation of neuronal genes during GNP differentiation.** (**a**) qRT–PCR analysis of Chd7 transcripts in P7 Chd7 WT [Chd7$^{f/f}$], heterozygous [Atoh1-Cre::Chd7$^{f/+}$] and homozygous [Atoh1-Cre::Chd7$^{f/f}$] mutant GNPs. The level of Gapdh transcripts was used for normalization. Bars represent normalized mean value ± s.d. from three samples for each group. Paired two-tailed t-test with equal variance was performed, $P = 0.0034$ (for Het.); $P = 0.001$ (for Hom.). (**b**) Heatmap shows significantly downregulated genes ($P < 0.01$, the Kruskal–Wallis test) in Chd7 mutant GNPs compared to WT. The colour scale is shown on the right. (**c**) DAVID Gene Ontology Biological Process analysis of genes downregulated in Chd7 homozygous mutant GNPs compared to WT. GO terms with Benjamini adjusted $P$-value $< 0.05$ are shown. (**d**) Heatmap exhibits significantly downregulated genes that are involved in neural function, in Chd7 mutant GNPs compared to WT. The colour scale is shown on the right. (**e**) ChIP-seq density heatmaps for Chd7 and H3K27ac within ± 5 kb of the Chd7 peak centre. (**f**) ChIP-seq density heatmaps show the enriched binding of Chd7 to super-enhancers (SEs) defined by H3K27ac. The start (S) and end (E) of SEs and ± 5 kb surrounding the enhancer regions are shown. (**g**) IGV track view of ChIP-seq density profile for Chd7 and H3K27ac of Neurod1 and Cadps2. The SE region is highlighted with red line. (**h**) IGV track view of ATAC-seq density profile (normalized to per million reads for each sample) at Cadps2 and Nkain3. The decreased ATAC-seq peaks are highlighted with grey rectangles.

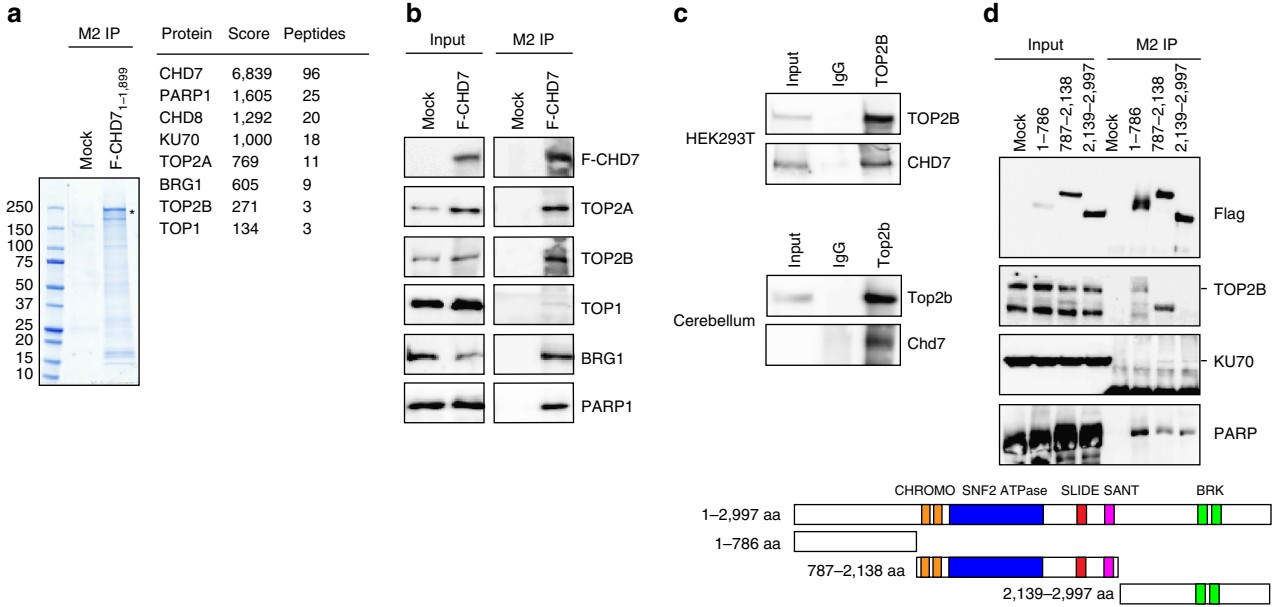

**Figure 5 | Chd7 interacts with Top2b.** (**a**) Coomassie-blue staining of an SDS gel shows immunoprecipitation using M2 beads in *CHD7* KO HEK293T cells overexpressing a Flag-tagged truncated CHD7$_{1-1,899}$ (highlighted with asterisk). Cells transfected with an empty vector served as mock control. Numbers on the left side indicate the molecular weight. A list of proteins identified by mass spectrometry, peptides of which were exclusively present in CHD7-immunoprecipitation but not mock control, is shown in the right panel. Scores and numbers of unique peptides identified for each protein are shown. (**b**) Immunoblotting of CHD7-interacting proteins precipitated with M2 beads in nuclear extracts from HEK293T cells overexpressing a Flag-tagged full-length CHD7 (F-CHD7) or mock-transfected cells. Forty per cent of IP and 2.5% of the input are shown. (**c**) Co-immunoprecipitation assay shows the interaction between endogenous Chd7 and Top2b in HEK293T and adult (6-week-old) mouse cerebella. Cell lysates were immunoprecipitated with an antibody against Top2b, and co-precipitated Chd7 was visualized by western blots. Five per cent of input and 100% of IP are shown. Note that endogenous Chd7 in adult cerebella was only detected by western blot after enrichment by immunoprecipitation. (**d**) The N-terminal part of CHD7 interacts with TOP2B. Cell lysates from HEK293T cells transfected with a series of constructs encoding Flag-tagged truncated CHD7 proteins were precipitated with M2 beads, and precipitated proteins were visualized by western blots. Fifty per cent of IP and 4% of the input are shown. Bands corresponding to TOP2B and KU70 according to their molecular weight are pointed with lines. The schemes illustrate the domain structure of full-length and truncated CHD7 protein used in the experiments.

23.2% of Top2b binding sites exhibited alteration of the association of Top2b (Fig. 6g). GO analysis of genes with decreased Top2b ChIP-seq peaks revealed cell morphogenesis, cell adhesion, ion transport and neuron differentiation as the most enriched biological process (Supplementary Fig. 9g), while no significantly (Benjamini adjusted $P$ value $< 0.05$) enriched term was detected in genes with increased peaks in *Chd7*-deficient CGNs. For instance, the binding of Top2b to promoters of two Chd7- and Top2b-activated genes *Cadps2* and *Cnksr3* was decreased in *Chd7* mutant CGNs compared to WT, whereas the binding of Top2b to its known binding site at the *Fos* promoter was not altered (Fig. 6h, Supplementary Fig. 9h). Importantly, GSEA revealed that genes carrying decreased Top2b ChIP-seq peaks in *Chd7*-deficient CGNs tended to be downregulated upon ICRF-193 treatment (Fig. 6i), revealing that Chd7-dependent recruitment of Top2b is necessary for the expression of these genes. Furthermore, while the average length of genes bound by Top2b (6147 genes) in WT CGNs was around 95 kb, the average gene length of genes carrying decreased Top2b binding peak (1190 genes) or increased Top2b binding peak (718 genes) upon loss of *Chd7* was around 153 or 115 kb, respectively. Thus, Chd7-dependent targeting of Top2b is preferentially observed in long genes. Moreover, we measured the amount of Top2b covalent complexes in *Chd7* WT and mutant CGNs treated with etoposide, a DNA topoisomerase poison that traps DNA with topoisomerases as a complex. We observed reproducible decrease, although not statistically significant ($P = 0.2169$), of Top2b covalent complexes in *Chd7*-deficient cells (Supplementary

Fig. 9i), supporting that Chd7 could contribute to the targeting of Top2b.

**Top2b regulates cerebellar development.** While the reelin signalling is essential for the proper localization of Purkinje cells[27], the cerebellar phenotype of *Top2b* knockout mice has not been reported so far. To investigate this, we made use of the CRISPR-Cas9 system to ablate *Top2b* in the developing cerebellum. The cerebella of E13.5 *Rosa26-CAG-LSL-Cas9-P2A-EGFP* embryos were electroporated with plasmids encoding *Cre recombinase* and either sgRNA against *Top2b* or a negative-control sequence. The transfection of cerebellar cells was efficient as shown at P7 by the wide expression of GFP upon Cre-mediated recombination (Fig. 7a). In both control and *Top2b*-targeted cerebella, most of GFP$^+$ cells in the IGL are Pax6$^+$ (Fig. 7b), demonstrating efficient targeting of CGNs by electroporation. Around 80% of GFP$^+$ cells in IGL of *Top2b*-targeted cerebella lost the expression of Top2b (Fig. 7b), demonstrating high knockout efficiency by the CRISPR-Cas9 system *in vivo*. Consistent with the data mentioned above (Fig. 6d), we observed that about 40% of cells lost Chd7 expression within transfected (GFP$^+$) cells upon targeting of *Top2b* (Fig. 7b). Furthermore, Top2b loss resulted in the downregulation of Zfpm2, a common target gene of Chd7 and Top2b (see Supplementary Data 1) (Fig. 7b). These data provide *in vivo* evidence that Top2b is required for the expression of neuronal genes in granule cells. Importantly, clusters of mislocalized Purkinje cells were observed in all three *Top2b*-targeted, but not in the two control cerebella

(Fig. 7c), as observed in *Chd7*-deficient [*Atoh1-Cre::Chd7^{f/f}*] mice. On the other hand, the percentage of proliferating (Ki67$^+$) and differentiating (p27$^+$) cells among GFP$^+$ cells in EGL were not altered upon mutation of *Top2b* (Fig. 7d), similar to our previous finding that *Chd7* loss does not affect cell proliferation of GNPs. Thus, the cerebella bearing *Chd7*- or *Top2b*-deficient granule cells partly share similar phenotype, which supports our conclusion

that expression of a subset of neuronal genes are regulated by Chd7 and Top2b.

## Discussion

Many CHARGE patients carrying *CHD7* mutation exhibit cerebellar hypoplasia and massive Purkinje cell heterotopia[4,19].

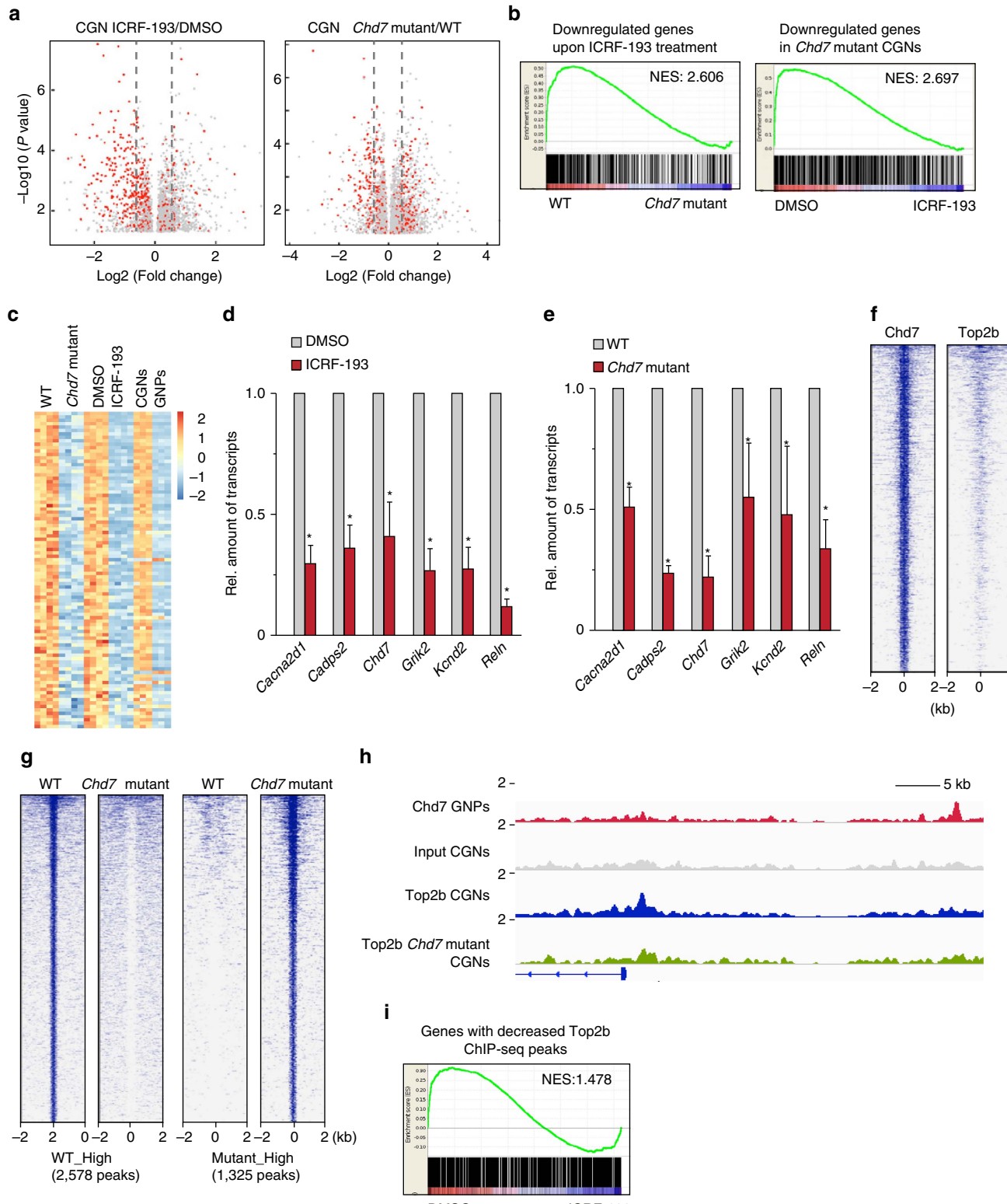

In both rodents and human, cerebellar development extends a long period from early embryonic stage until the first postnatal weeks (in mouse) or years (in human). One recent study revealed that Chd7 is necessary for cerebellum specification during early embryogenesis (around E8.5 in mouse) using *Chd7* gene trap mice[4]. However, the contribution of Chd7 during later cerebellar development and its implication in CHARGE syndrome has not been studied, however. In this study, cell type-specific inactivation of the *Chd7* gene using multiple Cre drivers identified failure of cerebellar granule cells differentiation as a contributing factor for cerebellar hypoplasia and Purkinje cell heterotopia. RNA-seq analysis revealed impaired downregulation of Reln in *Chd7*-deficient GNPs, by which mislocalization of Purkinje cells could be explained as observed in *reeler* mice[21,48,49]. Mechanistically, Chd7 is specifically associated with and is required for the activation of a core neuronal differentiation programme through maintaining the open chromatin structure and recruiting DNA topoisomerase IIb to these genes. Our data provide a clear epigenetic mechanism of Chd7-mediated gene transcriptional regulation *in vivo*, thus shedding light on epigenetic mutation-associated human diseases.

Our previous study revealed that loss of *Chd7* in adult neural stem cells impaired the transcriptional activation of two essential neurogenic transcription factors *Sox4* and *Sox11*, leading to defects of both the quantity and quality of newborn neurons[12]. In the current study, we made use of the abundance of primary cerebellar granule cells to perform comprehensive molecular analyses. Expression profiling data clearly revealed that Chd7 is required for the activation of genes involved in neural functions. ChIP-seq on freshly isolated GNPs demonstrated that Chd7 occupies active chromatin regions in genes involved in transcription and neuronal functions. Consistent with the result from human embryonic stem cells[31], Chd7 was found to be associated with SEs, which exhibit densely spaced clusters of active histone marks H3K27ac and clustered occupancy of cell identity-specific transcription factors. The binding of these transcription factors forming self-reinforcing circuits within SEs shapes cell-type-specific transcriptional landscape and thus governs cell identity. Interestingly, two SEs were identified within the *Chd7* gene, co-occupied by Chd7 protein and H3K27ac (Supplementary Fig. 5b), implicating that Chd7 could be a key chromatin regulator of cell identity in GNPs. In accordance with our study, a very recent investigation revealed an essential role of Chd7 in the onset of myelination via targeting the enhancers of key myelinogenic genes[30].

Both chromatin remodellers and DNA topoisomerases are crucial enzymes involved in DNA-related processes. The cooperative action of these two groups of enzymes was not appreciated until very recently[50]. In this study, we revealed

the collaboration between Chd7 and DNA topoisomerases Top2b to activate the expression of long neuronal genes. As an example, our molecular analyses identified *Reln* as a common target of both Chd7 and Top2b. Importantly, depletion of *Top2b* using *Foxg1-Cre*, a forebrain-specific Cre driver, leads to aberrant lamination of cerebral cortex due to the repression of *Reln* (ref. 39). Here, we demonstrated that the enzymatic activity of Top2b is crucial for the expression of Reln in cerebellar granule cells and genetic ablation of Top2b in cerebellum during development leads to mislocalization of Purkinje cells that is very likely due to lack of reelin signalling (Fig. 6c). In cortical neurons, the enzymatic activities of Top2b and Top1 are particularly necessary for transcription of long genes[38]. It has been thought that transcription of very long genes causes more supercoiled DNA double helix structures on these genes, thus requiring Top enzymes to solve this topological issue during transcriptional elongation. Our results suggest that besides DNA topoisomerases, chromatin-remodelling activity is critical for transcription of very long genes. Two recent studies have revealed the cooperation between another chromatin remodeller the BAF complex with DNA topoisomerases. The first study demonstrated that in dividing cells, the BAF complex interacts with Top2a and regulates Top2a-mediated decatenation activity during mitosis[50]. The second study revealed that the BAF complex-dependent chromatin remodelling activity is required for the Top1-mediated gene activation during inflammation[51]. Here, we observed Chd7 interacts with several topoisomerases including Top1, Top2a and Top2b (Fig. 5a). It will be of great interest to explore the potential functional interaction between Chd7 and Top1 in postmitotic neurons and Top2a in dividing cells. Furthermore, as reported in a previous study[34] and our current study (Fig. 5a), CHD7 protein interacts with the BAF complex. It could be interesting to investigate the cooperation action among CHD7, BAF complex and DNA topoisomerase.

The rapid progress of genomic sequencing technology has provided novel insights into genetic causes of many human diseases. On the other hand, the causal role of most of these newly identified mutations needs to be experimentally validated. It is striking that most of the patients carrying *CHD7* mutations display neurological disabilities, suggesting that CHD7 is a master chromatin regulator for brain function. Using mouse genetics, we here discovered that Chd7 controls a core neuronal differentiation programme during cerebellar granule cell differentiation. Importantly, 16 out of the 151 genes ($P = 3.146E-15$, Chi-square test) that were significantly downregulated in *Chd7*-deficient GNPs (Fig. 4b, Supplementary Data 1) were previously found as genetic mutation(s) in ASD patients (http://www.sfari.org). Furthermore, among 240 identified Chd7 and Top2b common activated genes in CGNs (Fig. 6c, Supplementary Data 1),

**Figure 6 | Chd7 recruits Top2b to activate long neuronal genes.** (**a**) Volcano plots depict gene expression changes ($P < 0.05$) between CGNs treated with ICRF-193 or DMSO (left panel) and between WT [*Chd7^f/f*] and *Chd7* homozygous mutant [*Atoh1-Cre::Chd7^f/f*] CGNs (right panel). Genes longer than 100 kb are labelled as red dots. The grey lines show the position of fold change of 1.5. (**b**) Gene Set Enrichment Analysis (GSEA) of genes that were significantly downregulated ($P < 0.01$) in CGNs upon ICRF-193 treatment in WT and *Chd7* mutant CGNs (left panel) and were significantly downregulated in Chd7 mutant CGNs ($P < 0.01$) in ICRF-193- and DMSO-treated WT CGNs (right panel). $P < 0.001$. NES: Normalized Enrichment Score. (**c**) Heatmap of the expression of genes activated by both Chd7 and Top2b in WT and *Chd7* mutant CGNs, and ICRF-193- or DMSO-treated CGNs and cells cultured with (GNPs) or without SAG (CGNs). The colour scale is shown on the right. (**d**) qRT–PCR analysis represented Top2b-activated genes in CGNs. The level of *Gapdh* transcripts was used for normalization. Bars represent normalized mean value ± s.d. from four samples for each group. $P = 0.0003; 0.001; 0.0035; 0.005; 0.0006; 1.0E-5$ (from left to right). (**e**) qRT–PCR analysis of the represented genes in *Chd7* WT and mutant CGNs. The level of *Gapdh* transcripts was used for normalization. Bars represent normalized mean value ± s.d. from four samples for each group. $P = 0.001; 1.8E-5; 0.0004; 0.028; 0.035; 0.0016$ (from left to right). (**f**) ChIP-seq density heatmaps for Chd7 in GNPs and Top2b in CGNs within ± 2 kb of the Chd7 peak centre. (**g**) ChIP-seq density heatmaps depict altered Top2b ChIP-seq peaks ($P < 0.01$, fold change > 2) between WT and *Chd7* mutant CGNs. Regions within ± 2 kb of the Top2b peak centre are shown. (**h**) Track view of ChIP-seq density profile for Chd7 in GNPs and Top2b in WT and *Chd7* mutant CGNs in the region upstream of *Cadps2* gene. (**i**) GSEA of selected genes that have at least one decreased Top2b ChIP-seq peak in *Chd7*-deficient CGNs in ICRF-193- and DMSO-treated WT CGNs. $P < 0.001$.

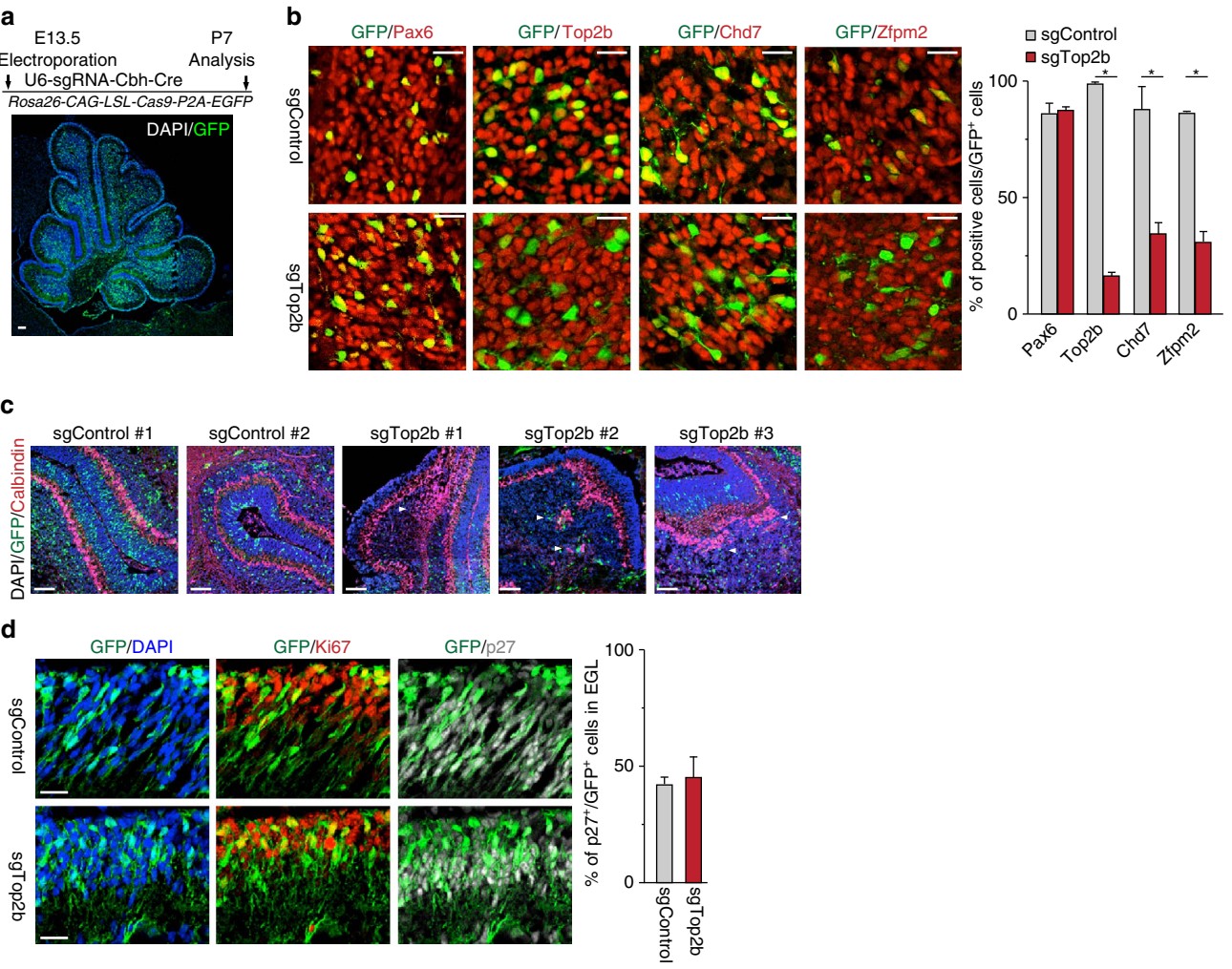

**Figure 7 | Genetic ablation of *Top2b* leads to phenotypic changes analogous to *Chd7*-deficient cerebella.** (**a**) Immunostaining of GFP (green) on P7 cerebellum from *Rosa26-CAG-LSL-Cas9-P2A-EGFP* mouse that was electroporated at E13.5 with a plasmid carrying sgRNA and Cre. Section is counterstained with DAPI (blue). Scale bar, 100 μm. The experimental scheme is shown in the top panel. (**b**) Immunostaining of GFP and Pax6, Top2b, Chd7 and Zfpm2 on P7 cerebella from *Rosa26-CAG-LSL-Cas9-P2A-EGFP* mouse that was electroporated at E13.5 with a plasmid construct expressing Cre and sgRNA against *Top2b* (sgTop2b) and a control sequence (sgControl). Quantification of positive cells among GFP$^+$ cells is shown in the right panel. Two and three pups were analysed for sgControl and sgTop2b, respectively. $P = 8.567E\text{-}06$ (for Top2b), $P = 0.0037$ (for Chd7), $P = 0.00046$ (for Zfpm2). Scale bars, 20 μm. (**c**) Immunostaining of GFP (green) and Calbindin (red) of P7 sgControl- and sgTop2b-electroporated cerebella. Arrowheads highlight mislocalized Purkinje cells. Scale bars, 100 μm. (**d**) Immunostaining of GFP (green), Ki67 (red) and p27 (white) of the EGL from P7 sgControl- and sgTop2b-electroporated animals. Quantification of p27$^+$ cells among GFP$^+$ cells was shown in the right panel. Scale bars, 20 μm.

mutations of 35 genes are associated with ASD ($P < 2.2E\text{-}16$, Chi-square test). Emerging evidence has linked the cerebellum with higher cognitive functions. Indeed, cerebellar deficits have long been implicated in autism[52]. Our study provides supporting evidence showing Chd7-dependent regulation of transcription is involved in the pathogenesis of autism. Moreover, many Chd7-regulated genes identified in this study, such as *Gap43*, *Cadps2* and *Reln*, function in other brain regions besides cerebellum[21,26,53]. Thus, the molecular network controlled by Chd7 maybe a core network for brain development and function in human.

## Methods

**Animals.** *Chd7* conditional knockout mice were generated by breeding of *Chd7^{f/f}* (ref. 12) with *Nestin-Cre* (JAX, #003771), *Atoh1-Cre* (JAX, #011104) and *Ptf1a-Cre* mice (BCBC, RES184). *Chd7-GFP* (from the GENSAT project, http://www.gensat.org/index.html) and *Rosa26-CAG-LSL-Cas9-P2A-EGFP* (JAX, #024857) mice were purchased. Mice were housed in a vivarium with

a 12-h light/dark cycle. All mice were bred in C57BL/6N background. Genotyping primers are listed in Supplementary Table 2. For *in utero* electroporation experiments, *Rosa26-CAG-LSL-Cas9-P2A-EGFP* homozygous knockin male mice were crossed with CD-1 female mice, yielding embryos carrying *Rosa26-CAG-LSL-Cas9-P2A-EGFP* heterozygous genotypes. For *in vivo* BrdU or IdU labelling, animals were subjected to intraperitoneal injection of BrdU or IdU (3 mg per 40 g body weight) and were killed at defined stages. All animal experiments were conducted according to animal welfare regulations and have been approved by the responsible authorities (Regierungspräsidium Karlsruhe, approval numbers: G32/14, G48/14 and G90/13).

**Plasmids.** The human CHD7 (1–1,899) cDNA with a C-terminal Flag tag was amplified from pFastbac-CHD7 (1-1,899) (a gift from Robert Kingston)[32], and cloned into pEF-DEST51 using the Gateway technology (Thermo Fisher Scientific). The full-length CHD7 expression construct containing an N-terminal GFP tag was inserted into the Flag-tagged CHD7-encoding expression plasmid[34]. The truncated CHD7 constructs were generated by the insertion of PCR products into pcDNA3.1 empty vector. The coding sequence of sgRNA against *CHD7* (CTTTCTAGAGAAACCAGTGC) was cloned into pLenti-CRISPR v2 (Addgene #52961). For genetic ablation of *Top2b*, the coding sequence of sgRNA against *Top2*b (target sequence: #1: CTTCGTCCTGATACATACAT; #2:

ACTGATCCAATGTATGTATC) or control region (target sequence: GCGACCA ATACGCGAACGTC) was cloned into the pU6-sgRNA-Cbh-Cre plasmid, which was modified from pX330 (Addgene #42230) by replacing the *hSpCas9* with *Cre* coding sequence.

**In utero electroporation.** *In utero* electroporation was performed as described previously[54] with some modifications. To ablate *Top2b*, pU6-sgRNA-Cbh-Cre $(2\,\mu g\,ml^{-1})$ was injected into the fourth ventricle and electroporated (35 V, 50 ms-on, 950 ms-off, 5 pulses) into the cerebellar primordial of E13.5 *Rosa26-CAG-LSL-Cas9-P2A-EGFP* embryos. Only GFP-labelled cerebella were used for further analyses. No specific randomization or blinding was performed.

**Primary cerebellar granule cell preparation.** GNPs were purified from P7 C57BL/6N pups using percoll gradient centrifugation and plated on matrigel-coated plates as described previously[24]. We did not discriminate the gender of pups. For BrdU analysis, freshly isolated cells were analysed by using the flow cytometry after staining with an anti-BrdU antibody (BD Pharmingen). In some experiments, the cells were cultured for defined days with or without either Shh protein or SAG (200 nM) and collected for analysis. To inhibit Top2b activity, cells were treated with ICRF-193 (Sigma-Aldrich) in DMSO at $10\,\mu M$ for 12 h before collecting them. To delete *Chd7 in vitro*, *Chd7 f/f* GNPs were cultured without Shh, but treated with TAT-Cre (Merck Millipore) at $0.5\,\mu M$ for 48 h before collecting them for RNA analysis.

**Culture and transfection of HEK293T cells.** Mycoplasma-free HEK293T cells from American Type Culture Collection were cultured in DMEM-Glutamax, (ThermoFisher Scientific) supplemented with 10% fetal bovine serum (ThermoFisher Scientific), $100\,U\,ml^{-1}$ of penicillin-streptomycin (ThermoFisher Scientific). For transient transfection, cells were transfected with a DNA-PEI (Sigma-Aldrich) mixture and collected 48 h later. To establish *CHD7* knockout cell line with the CRISPR-Cas9 technology, HEK293T cells were transfected with pLentiCRISPR-sg*CHD7* plasmid, selected with puromycin $(1\,\mu g\,ml^{-1})$ for 7 days before single cell clones were picked up and analysed.

**Preparation of cerebellar lysates.** Cerebella from adult (2-month-old) mice were freshly dissected. After snap freezing in liquid nitrogen, tissue was thawed on ice. Cerebella were homogenized in buffer A (10 mM HEPES, pH 7.9, 10 mM KCl, 1.5 mM $MgCl_2$), and incubated on ice for 30 min. After centrifugation, nuclei were dissolved in IP buffer (20 mM Tris-HCl, pH 8.0, 150 mM NaCl, 1 mM EDTA, 1 mM EGTA, 1% Triton X-100, complete protease inhibitor cocktail (Roche)).

**Immunoprecipitation and mass-spectrometry.** We performed IP as described previously[55]. Briefly, nuclei from transfected HEK293T were lysed in IP buffer, and treated with Benzonase (Sigma-Aldrich, $700\,U\,ml^{-1}$) for 1 h at 4 °C. Nuclear extracts were incubated at 4 °C overnight with M2 agarose (Sigma-Aldrich) or protein A/G sepharose (GE healthcare) with primary antibodies. After successive washing with IP buffer containing 200 mM NaCl, precipitated proteins were eluted in Laemmli buffer and analysed on western blots. Assays were repeated at least three times and the representative images were selected. For MS analysis, precipitated proteins were eluted with $400\,ng\,\mu l^{-1}$ 3 × Flag peptides (Sigma-Aldrich) in IP buffer containing 300 mM NaCl. Proteins were digested in solution with trypsin, and tryptic peptides were analysed with the Orbitrap XL mass spectrometer. The peptides were identified with the MASCOT searching engine against the SwissProt database. Uncropped immunoblots are shown in Supplementary Fig. 10.

**Immunostaining.** IHC was performed essentially described previously[24]. The animals were killed at defined stages and fixed with 4% paraformaldehyde in PBS (PFA/PBS) overnight, followed by sectioning with cryostat (Leica) and microtome (Leica). For immunofluorescence on cultured GNPs, cells were fixed with 4% PFA/PBS for 10 min at room temperature, permeabilized for 5 min in PBS containing 0.2% Triton X-100. For BrdU staining, sections were treated with 2 N HCl for 30 min at 37 °C. Tissue sections and cultured cells were subsequently subjected to the primary antibody incubation for overnight at 4 °C after blocking with 10% normal donkey serum. For IHC, paraffin sections were incubated with biotinylated secondary antibodies (Vector) and signals were amplified by a horseradish peroxidase system (ABC kit, Vector) followed by DAB staining (Sigma-Aldrich). Fluorescent images were captured using a confocal laser-scanning microscopy (LSM780 and 800, Zeiss; and SP5, Leica). At least eight sections from one mouse were stained. Cell counting was done in a double-blind manner whenever applicable. Representative images were selected from sections from at least three independent mice.

**Antibodies.** Antibodies used in this study were the following: BrdU (1:500, AbDserotec, OBT0030CX), Brg1 (1:200, Santa Cruz Biotechnology, sc-374197), Cadps2 (1:500, Merck Millipore, ABN326), Calbindin (1:500, Swant, CB38), Chd7 (1:400, Cell Signaling, #6505), cleaved Caspase3 (1:200, Cell Signaling, #9661),

Dab1 (1:200, Sigma-Aldrich, Ab232), Gap43 (1:500, Sigma-Aldrich, G9264), Ki67 (1:1,000, Abcam, ab15580), Ku70 (1:500, ThermoFisher Scientific, MA5-13110), GFP (1:500, Life technologies A11122; Abcam ab13970), H3K27Ac (2 μg per ChIP, Diagenode, pAb-174-050), NeuN (1:500, Merck Millipore, MAB377), p27 (1:500, BD bioscience, 610241), Parp (1:500, Cell Signaling,), Pax2 (1:1,000, Life technologies, 71-6000), Pax6 (1:500, Covance, PRB-278P), Reln (1:200, Merck Millipore, MAB5366), RPA116 (ref. 54), Sox2 (1:200, Santa Cruz Biotechnology, sc17320), Top2a (1:500, Santa Cruz Biotechnology, sc3659), Top2b (1:500, Santa Cruz Biotechnology, sc13059) and Zfpm2 (1:200, Santa Cruz Biotechnology, sc10755). These antibodies were quality-checked in previous publications.

**Quantitative RT–PCR (qRT–PCR).** Total RNAs were extracted from cultured GNPs and tumour tissues with TRI-Reagent (Sigma-Aldrich) and purified with the RNeasy Mini kit (Qiagen). RNA was transcribed into cDNA using random primers (dN6, Roche) and M-MLV reverse transcriptase (Promega). cDNA were quantified by using SYBR gene expression assays (Qiagen) or Taqman Probe with Absolute Blue qPCR Rox mix (ThermoFisher Scientific), on the CFX96 Real-time System (Bio-rad). Primers and Probes used for qRT–PCR were listed in Supplementary Table 2.

**Chromatin immunoprecipitation.** We performed the IP as described previously[54]. Freshly isolated P7 GNPs were fixed with 1% formaldehyde for 10 min at room temperature, and quenched with 125 mM glycine. Chromatin was extracted and sonicated for 20 min (30 s on, 30 s off) using a sonicator (Diagenode). After pre-clearing for 1 h by incubating with protein A/G beads, chromatin lysate was incubated overnight at 4 °C with antibodies. Beads were extensively washed. DNA–protein complex was eluted, reverse cross-linking and purified using Qiagen PCR purification column.

**Topoisomerase covalent complex assay.** Samples were processed according to the *in vivo* complex of enzyme (ICE) assay CGNs (ref. 56). CGNs were treated with etoposide (100 μM) for 1 h at 37 °C, and lysed in 1% sarkosyl in TE (10 mM Tris HCl, pH 7.5; 1 mM EDTA). DNA was sheared by passing the lysate ten times through a 2 ml syringe with a 25G/8 Gauge needle. Samples were centrifuged at 57,000 r.p.m. for 20 h at 25 °C using 3.3 ml 13 × 33 mm polyallomer Optiseal tubes (Beckman Coulter) in a TLN100 rotor (Beckman Coulter). Precipitated DNA (5 μg) was transferred onto Hybond ECL membranes (GE Healthcare) using a Bio-Dot SF Microfiltration Apparatus (Bio-Rad). Membranes were blocked 1 h with Odyssey Blocking Buffer (LI-COR Biosciences), incubated with anti-TOP2b antibody (1:1,000 dilution), in Odyssey Blocking Buffer-0.1% Tween20, washed (three times with TBS-0.1% Tween20), incubated with 1/15,000 IRDye 800CW Goat anti-Rabbit IgG (LI-COR) and finally washed (three times with TBS-0.1%Tween20 and once with TBS). Dried blots were analysed and quantified in Odyssey CLx using ImageStudio Odyssey CLx Software.

**Gene expression analysis.** Total RNA was extracted and subsequently profiled using the Affymetrix microarray chip (430V2). For RNA-sequencing, a sequencing library was prepared by using the Illumila TrueSeq Total RNA Sample Prep Kit and sequenced on the Illumina Hi-Seq 2000, single-end 50 bp reads. For each sample, about 20,000,000 single-end reads were obtained. RSEM-1.2.15 with Bowtie-1.0.0 is used to map the reads to the Ensemble mouse reference genome assembly mm10 transcriptome, and to estimate the expression levels for all the genes. For both Bowtie and RSEM, default parameters were used. Transcripts per million reads were adopted to quantify gene expression levels. RNA samples were sequenced in two batches, with the following samples as the first batch, *Chd7* WT-1, 2, 3; *Chd7* heterozygous mutant-1, 2, 3 and *Chd7* homozygous mutant-1, 2, and the rest samples as the second batch. For this, we used ComBat with default parameters to remove the batch effect[57].

**ATAC-seq analysis.** ATAC-seq was performed with minor modification according to the original method[33]. Briefly, 100,000 GNPs from P7 pups were isolated, permeabilized and incubated with Tn5 transposase (Illumina) for 30 min at 37 °C. DNA from Tn5-treated cells was purified and used as a template to amplify a DNA library by PCR. The Libraries were deep-sequenced with HiSeq 2000 v4 paired-end 125 bp. The ATAC-seq analysis was performed according to published papers[33,58]. There are three technical replicates for both *Chd7* WT and homozygous mutant GNPs, and two replicates for *Chd7* heterozygous mutant GNPs. For each sample, approximately 30,000,000 paired-end reads were sequenced. Reads were first trimmed with trim-galore_0.4.1 to remove the adapters and trimmed reads are mapped with bowtie 1.0.0[1] to mm10 with parameters -X 2000 -m 1. Duplicated reads and reads from mitochondrial DNA were removed with picard_1.139. To call the peaks, macs2 2.1.0.20150731 (ref. 59) were applied with parameters -nomodel - shift -100 -extsize 200. In order to calculate total amount of peaks, peaks given by macs2 from each of above-mentioned three genotypes were merged when peak distance is less than 300 b.p. Differential peaks were identified with edgeR (ref. 60) according to the number of reads mapped to each peak counted by Rsubread (refs 61,62) for each replicate. Both edgeR and Rsubread were run with default

parameters. ATAC-seq peaks are annotated using PAVIS (ref. 61) with default parameters. Coverage density plots were done with deeptools.

**Analysis of ChIP-seq.** About 5 million GNPs collected from P7 pups were used for H3K27ac ChIP-seq according to standard ChIP protocol described above. DNA libraries generated from ChIP-DNA and input-DNA were deep-sequenced using HiSeq pair-end 125 bp. After trimming with trim_galore, more than 40 million pair-end reads were mapped to mm10 genome with bowtie 1.0.0 with parameter -X 1000 -m 1. Peaks were called with macs2 with default parameters for Chd7 and H3K27ac ChIP-seq on GNPs. For Top2b ChIP-seq (WT and *Chd7* mutant CGNs), we use homer[58] findPeaks with parameter -F 2.0 -P 0.001 -L 2.0 -LP 0.001 -fdr 0.01 -style factor. For the differential peaks of Top2b Chip seq, MAnorm[59] was used with cutoff $P < 0.01$ and fold change $> 2$. SE was called using homer findPeaks with parameter -style super. ChIP-seq peaks are annotated using PAVIS (ref. 61) with default parameters. Coverage density plots were done with deeptools.

**GO analysis and GSEA.** Gene ontology biological process was performed with David6.7 (http://david.ncifcrf.gov). GO term with the Benjamini adjusted $P$ value $< 0.05$ is considered to be significantly enriched. For custom gene sets, GSEA (www.broadinstitute.org/gsea) was performed using default parameter. FDR $< 25\%$, $P < 0.05$ was considered to be significantly associated.

**Statistical methods.** Two-tailed Student's *t*-test with equal variance was performed to compare two groups. The Kruskal − Wallis test was performed for tests with three groups. Pearson test and Chi-squared test were performed as mentioned. For significance test, *means $P < 0.05$. Error bar stands for mean ± s.d.

**Data availability.** All sequencing and microarray data have been submitted to the US National Institute of Health GEO database (https://www.ncbi.nlm.nih.gov/gds) under the accession number GSE93741.

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

## Acknowledgements

We thank Drs R. Kingston and H. van Attikum for providing us the *CHD7* expression constructs; C. Shao, L. Linke and N. Mack for excellent technical assistance, Drs A. Forget, K. Reifenberg, N. Denk and K. Dell for helpful assistance for animal experiments; the Imaging and Cytometry, Genomics and Proteomics Core Facilities of the DKFZ and the Carl Zeiss Imaging Center in the DKFZ for their support. This work was supported by the Helmholtz Association (VH-NG-702), the Deutsche Forschungsgemeinschaft (KA 4472/1-1 for D.K., LI 2140/1-1 for H.-K.L.), the Deutsche Krebshilfe (110226 to H.-K.L.), the ERC (European Research Council) consolidator grant (647055) (to H.-K.L.), the Helmholtz Alliance 'Preclinical Comprehensive Cancer Center' Grant HA-305 (to J.G., P.L. S.M.P. and H.-K.L.), the DKFZ Intramural Grant (to W.F. and D.K.), the Spanish Government (SAF2010-21017, SAF2013-47343-P, SAF2014-55532-R and FEDER funds, to F.C.-L.), the Andalusian Regional Government (P11-CVI-7948 and FEDER funds, to F.C.-L.), the European Research Council (ERC-CoG-2014-647359, to F.C.-L.) and Predoctoral Studentships from the University of Seville to J.A.L. (PIF-2011). CABIMER is supported by the Andalusian Regional Government (Junta de Andalucía).

## Author contributions

W.F., D.K., S.M.P and H.-K.L. conceived the project. D.K., W.F., S.M.P. and H.-K.L. supervised the project. W.F., D.K., H.D., E.S., L.S., O.F., M.J., and A.N. analysed cerebellar tissues and primary granule cells. W.F. B.S.H. and E.S. performed experiments on cell lines. J.A.L. performed the Topoisomerase covalent complex assay. H-K.Q. analysed RNA-sequencing data, Affymetrix microarray, ATAC-seq data and ChIP-seq data. A.K. performed histopathological analysis. S.L., D.T.W.J., M.K. and P.A.N. analysed human data. Y.H., V.R., M.Z. and O.A. provided biomaterials. J.G., O.A. and P.L. provided intellectual inputs. W.F., D.K., J.A.L., J.G., D.T.W.J. M.K., F.C.-L., S.M.P. and H.-K.L. edited and wrote the manuscript.

## Additional information

**Competing interests:** The authors declare no competing financial interests.

