## [Peer Review File · Nature Communications]

Transferred manuscripts:

Reviewer expertise:

Reviewer #1: Transcriptional regulation in neurodevelopmental disorders;

Reviewer #2: CNS tumor;

Reviewer #3: Cerebellum development, transcriptional regulation.

REVIEWERS' COMMENTS:

Reviewer #1 (Remarks to the Author):

The authors have substantially improved the manuscript with the addition of new experiments and the inclusion of more in-depth analysis of the genomic datasets. This study will provide valuable data and insight to the field. I am happy to recommend publication.

Reviewer #2 (Remarks to the Author):

The authors have revised the manuscript to address many, but not all, of the original concerns. The authors have performed a large number of high quality studies. However, they often conclude that there is a causation when the data only involve an association. For example, the authors mass spec data show that CHD7 interacts with several other proteins, of which Top2b has one of the lower scores. The interaction between CHD7 and Top2B seems real, but it is possible that CHD7 interacts with something other than Top2B alone and the other protein could be responsible for directing Top2B binding. The targeting of CHD7 and Top2B supports that each is necessary, but the needed rescue and over expression studies to prove sufficiency are lacking. I would suggest that the authors can avoid most experiments, if they carefully and thoroughly revise the text of the manuscript and figure legends to avoid claims that extend beyond the scope of the data. One area that was not well done was the tumor data. Just because the data were moved to the Supplemental Data section, it doesn't mean that they can avoid doing the requested studies. I would recommend its removal or they should address the original requests much better. Overall, this is an interesting study with substantial amounts of quality data.

Statements that should be addressed (there are others):

Abstract: "Moreover, Chd7 interacts with DNA topoisomerase Top2b in cerebellar granule neurons to specifically activate very long genes, including Reln that is crucial for appropriate differentiation of Purkinje cells." The Reln data are only associations and there are no functional data to show that Reln is the mechanism by which Chd7 alters the Purkinje cells.

Abstract: "Besides these neurological phenotypes, genetic loss of Chd7 accelerates medulloblastoma formation in vivo by blocking neural progenitor differentiation." The change in differentiation was not shown to be the mechanism; there is only an association.

Page 8: "Chd7 is required for the expression of genes associated with neuronal differentiation." I would suggest rewriting this as the direct evidence is good, but not complete. They may want to state,

"Chd7 disruption is associated with reduced expression of genes associated with neuronal differentiation."

Page 8: "Notably, given that the reelin signaling is essential for the proper formation of the Purkinje cell layer, the observed phenotype of mislocalization of Purkinje cells in [Atoh1-Cre::Chd7 f/f] mice could be explained by impaired Reln expression in Chd7 mutant granule cells." This is speculative and should be removed from the Results. It could be moved to the Discussion.

Page 9: "Collectively, ChIP-seq and ATAC-seq data provides direct evidence demonstrating that Chd7 is associated with and required for the maintenance of open chromatin at the regulatory elements of genes involved in neuronal differentiation." The experiments used in support of this statement do not fully address this. The lack of rescue studies makes these claims premature.

"Importantly, upon loss of Chd7, 23.2% of Top2b binding sites exhibited alteration of the association of Top2b (Fig. 6g), demonstrating that Chd7 is required for the proper targeting of Top2b." The fact that most Top2b binding sites remain intact make this statement clearly overly generalized. There may be a subset or another interacting protein could be responsible.

Page 14, 15: "These data revealed that in Chd7 mutant-MBs the differentiation gene expression program was not properly executed, therefore Chd7 mutant-MBs is less differentiated compared to Chd7 WT MBs." The entire tumor section is still weak with very little additional data. The causation is not established. I would suggest that the authors remove these data and perform the needed experiments in another manuscript in which the role of Chd7 in medulloblastoma could be more rigorously addressed.

Reviewer #3 (Remarks to the Author):

This manuscript investigates the role of the chromodomain helicase DNA-binding protein 7 (Cdh7) in cerebellar development, and uses a combination of different sequencing methods to identify potential mechanisms by which CDH7 regulates chromatin and thus gene activation. The authors found that loss of Cdh7 causes cerebellar hypoplasia, impairs granule cell differentiation, induces apoptosis, and causes mis-localization of Purkinje cells. A combination of RNA-seq, ChIP-seq, and ATAC-seq suggests that CDH7 regulates chromatin to activate genes important for granule cell differentiation. They also showed that Chd7 interacts with the DNA topoisomerase Top2b to activate long genes, that inactivation of Top2b using CRISPR-Cas9 methodology also causes Purkinje cell mis-localization, and that loss of Cdh7 accelerates tumor formation, which is associated with a decrease in the expression of differentiation genes. While the cerebellar phenotype caused by loss of Cdh7 is interesting, and the sequencing data offer insight into the cellular processes and mechanisms affected, the findings are largely correlative. Although the authors attempted to extend these findings by investigating the interaction of CDH7 with Top2b and how loss of Cdh7 affects tumor formation, these final experiments are cursory and do not tie together or offer enough insight into how CDH7 and Top2b interact to regulate granule cell development and tumor formation. Thus, although the data presented in this manuscript are interesting and promising, more data are needed to fully define a particular pathway or mechanism responsible for the effects of loss of Cdh7 on cerebellar development, granule cell differentiation and tumor formation. As such, the manuscript is better suited for a more specialized journal.

Concerns:

For Figure 2a, the lateral hemispheres don't really look smaller (hypoplasia) pre and post-natally, as indicated in the text.

For Figure 2b, are some of the mutant sections more lateral compared to the control sections? Lobe 9, which looks normal for the mutant in the Nissl staining, probably because of less Cre expression in the posterior lobes, does not look typical of mid-sagittal sections in the immunostained sections.

For Figure 3b, the immunostained images look the same at that magnification, although a difference is indicated on the graph.

The authors should have counted the percentage of BrdU positive cells in the IGL for the 48 hr BrdU pulse to see how many cells make it to the IGL to compliment the NeuN Figure 3c.

The effects of down-regulation of Reln on Purkinje cell placement the Cdh7 mutant are purely speculative. There is no existing evidence for a critical role for RELN in PC alignment.

How specific is ICRF-193?

Need to indicate Figure 7d in the text.

Most of the heat maps shown by the authors do not include gene names listed down the side. Also, are there links to the original data for all of the sequencing experiments? These data should have been included in the manuscript.

The Top2b and tumor experiments at the end of the manuscript appear to be cursory and do not tie together. The authors should have investigated and reported the effects of loss of Top2b on granule cell development, not only on Purkinje cell displacement. Also does Top2b contribute to tumor formation? Why was the tumor formation data shuttled to the supplementary info? As a result, the paper seems to jump around a bit and lose focus at the end.

REVIEWERS' COMMENTS:

Reviewer #1 (Remarks to the Author):

The authors have substantially improved the manuscript with the addition of new experiments and the inclusion of more in-depth analysis of the genomic datasets. This study will provide valuable data and insight to the field. I am happy to recommend publication.

We are happy to satisfy the reviewer #1 with the revision.

Reviewer #2 (Remarks to the Author):

The authors have revised the manuscript to address many, but not all, of the original concerns. The authors have performed a large number of high quality studies. However, they often conclude that there is a causation when the data only involve an association. For example, the authors mass spec data show that CHD7 interacts with several other proteins, of which Top2b has one of the lower scores. The interaction between CHD7 and Top2B seems real, but it is possible that CHD7 interacts with something other than Top2B alone and the other protein could be responsible for directing Top2B binding. The targeting of CHD7 and Top2B supports that each is necessary, but the needed rescue and over expression studies to prove sufficiency are lacking. I would suggest that the authors can avoid most experiments, if they carefully and thoroughly revise the text of the manuscript and figure legends to avoid claims that extend beyond the scope of the data. One area that was not well done was the tumor data. Just because the data were moved to the Supplemental Data section, it doesn't mean that they can avoid doing the requested studies. I would recommend its removal or they should address the original requests much better. Overall, this is an interesting study with substantial amounts of quality data.

According to the reviewer #2's suggestions, we've carefully revised our statements (highlighted in the text) in the manuscript to avoid overstatement beyond actual data presented here. Regarding the tumor data, we admitted insufficiency of data showing contribution of Chd7 to tumor formation. Thus, we completely left out the tumor part in the revised.

Statements that should be addressed (there are others):

Abstract: "Moreover, Chd7 interacts with DNA topoisomerase Top2b in cerebellar granule neurons to specifically activate very long genes, including Reln that is crucial for appropriate differentiation of Purkinje cells." The Reln data are only associations and there are no functional data to show that Reln is the mechanism by which Chd7 alters the Purkinje cells.

We would mention in the ABSTRACT that known functionally important genes (e.g. Reln) are affected by dysregulation of Chd7 or Top2b expression. To avoid the misleading, we rephrased it based on our data obtained in this study (line 41-43).

Abstract: "Besides these neurological phenotypes, genetic loss of Chd7 accelerates medulloblastoma formation in vivo by blocking neural progenitor differentiation." The change in differentiation was not shown to be the mechanism; there is only an association.

According to the reviewers' suggestions, we completely removed the tumor part in this manuscript.

Page 8: "Chd7 is required for the expression of genes associated with neuronal differentiation." I would

suggest rewriting this as the direct evidence is good, but not complete. They may want to state, “Chd7 disruption is associated with reduced expression of genes associated with neuronal differentiation.”

According to the reviewer#2's suggestion, we rephrased it with the word limitation (line 180).

Page 8: “Notably, given that the reelin signaling is essential for the proper formation of the Purkinje cell layer, the observed phenotype of mislocalization of Purkinje cells in [Atoh1-Cre::Chd7 f/f] mice could be explained by impaired Reln expression in Chd7 mutant granule cells.” This is speculative and should be removed from the Results. It could be moved to the Discussion.

We removed the sentence in the RESULT and added one new sentence about relationship between the phenotype of Purkinje cell mislocalization and Reln loss in the DISCUSSION (line 368-369).

Page 9: “Collectively, ChIP-seq and ATAC-seq data provides direct evidence demonstrating that Chd7 is associated with and required for the maintenance of open chromatin at the regulatory elements of genes involved in neuronal differentiation.” The experiments used in support of this statement do not fully address this. The lack of rescue studies makes these claims premature.

We rephrased our statement (line 233-235).

“Importantly, upon loss of Chd7, 23.2% of Top2b binding sites exhibited alteration of the association of Top2b (Fig. 6g), demonstrating that Chd7 is required for the proper targeting of Top2b.” The fact that most Top2b binding sites remain intact make this statement clearly overly generalized. There may be a subset or another interacting protein could be responsible.

We reworded according to this suggestion (line 316-317; 329-330 and 334-335).

Page 14, 15: “These data revealed that in Chd7 mutant-MBs the differentiation gene expression program was not properly executed, therefore Chd7 mutant-MBs is less differentiated compared to Chd7 WT MBs.” The entire tumor section is still weak with very little additional data. The causation is not established. I would suggest that the authors remove these data and perform the needed experiments in another manuscript in which the role of Chd7 in medulloblastoma could be more rigorously addressed.

The tumor part was completely removed in the manuscript.

Reviewer #3 (Remarks to the Author):

This manuscript investigates the role of the chromodomain helicase DNA-binding protein 7 (Cdh7) in cerebellar development, and uses a combination of different sequencing methods to identify potential mechanisms by which CDH7 regulates chromatin and thus gene activation. The authors found that loss of Cdh7 causes cerebellar hypoplasia, impairs granule cell differentiation, induces apoptosis, and causes mis-localization of Purkinje cells. A combination of RNA-seq, ChIP-seq, and ATAC-seq suggests that CDH7 regulates chromatin to activate genes important for granule cell differentiation. They also showed that Chd7 interacts with the DNA topoisomerase Top2b to activate long genes, that inactivation of Top2b using CRISPR-Cas9 methodology also causes Purkinje cell mis-localization, and that loss of Cdh7 accelerates tumor formation, which is associated with a decrease in the expression of differentiation genes. While the cerebellar phenotype caused by loss of Cdh7 is interesting, and the sequencing data offer insight into the cellular processes and mechanisms affected, the findings are largely correlative. Although the authors attempted to extend these findings by investigating

the interaction of CDH7 with Top2b and how loss of Cdh7 affects tumor formation, these final experiments are cursory and do not tie together or offer enough insight into how CDH7 and Top2b interact to regulate granule cell development and tumor formation. Thus, although the data presented in this manuscript are interesting and promising, more data are needed to fully define a particular pathway or mechanism responsible for the effects of loss of Cdh7 on cerebellar development, granule cell differentiation and tumor formation. As such, the manuscript is better suited for a more specialized journal.

Concerns:

For Figure 2a, the lateral hemispheres don't really look smaller (hypoplasia) pre and post-natally, as indicated in the text.

We observed hypoplasia in the cerebellar vermis but not hemisphere of *Chd7*-deficient cerebella at P0 and P7. In adult mutant mice, hypoplasia was seen at both vermis and hemisphere. To make this clearer, we revised the sentence in the RESULTS (line 132-135). On the other hand, the foliation defects were observed in both vermis and hemisphere from P0 onwards.

For Figure 2b, are some of the mutant sections more lateral compared to the control sections? Lobe 9, which looks normal for the mutant in the Nissl staining, probably because of less Cre expression in the posterior lobes, does not look typical of mid-sagittal sections in the immunostained sections.

In Figure 2b, we replaced the original picture of the wild-type cerebellum with the new one showing at the similar medio-lateral level. Notably, mislocalized Purkinje cells were not observed in any control section.

For Figure 3b, the immunostained images look the same at that magnification, although a difference is indicated on the graph.

We apologize for the confusion. We replaced original pictures with higher magnification views in Figure 3b.

The authors should have counted the percentage of BrdU positive cells in the IGL for the 48 hr BrdU pulse to see how many cells make it to the IGL to compliment the NeuN Figure 3c.

Thanks for this good suggestion. We have counted the number of BrdU⁺ cells in IGL and observed the decrease of BrdU⁺ cells in *Chd7* mutant animals, supporting our conclusion that loss of Chd7 leads to less granule cells in IGL. The data is presented in Supplemental Fig. 3f.

The effects of down-regulation of Reln on Purkinje cell placement the *Cdh7* mutant are purely speculative. There is no existing evidence for a critical role for RELN in PC alignment.

We noticed that we have to make the previous findings clearer in the text. A few previous studies have revealed massive mislocalization of PCs in reeler mice. Also, one study has shown that in vivo functional blocking of Reelin with the neutralized CR-50 antibody caused migration-defects in PCs (Miyata et al., Journal of neuroscience, 1997). We cited these papers (line 369-370).

How specific is ICRF-193?

ICRF-193 inhibits both Top2a and Top2b. Because Top2a is repressed in cerebellar granule neurons (supplementary Fig. 8a), ICRF-193 specifically inhibits Top2b in these cells. We also cited the studies showing specificity of ICRF-193 (line 275).

Need to indicate Figure 7d in the text.

We apologize for our mistake. We fully described Figure 7d now (line 354-356).

Most of the heat maps shown by the authors do not include gene names listed down the side. Also, are there links to the original data for all of the sequencing experiments? These data should have been included in the manuscript.

We show the lists of genes presented in figure 2b and figure 6c as an excel file (supplementary Table S3).

The Top2b and tumor experiments at the end of the manuscript appear to be cursory and do not tie together. The authors should have investigated and reported the effects of loss of Top2b on granule cell development, not only on Purkinje cell displacement. Also does Top2b contribute to tumor formation? Why was the tumor formation data shuttled to the supplementary info? As a result, the paper seems to jump around a bit and lose focus at the end.

According to the suggestion from reviewers, tumor part has now been removed from the manuscript. To monitor the effect of *Top2b* loss in EGL, we performed co-immunostaining of GFP and two mutually exclusive markers Ki67 (proliferation) and p27 (differentiation). Quantification results show no difference of the ratio of p27⁺ cells among GFP⁺ cells between control and *Top2b* mutant (**Figure 7d**), indicating the proliferation and differentiation status of granule cells in EGL is not perturbed. In addition, we hardly detect any GFP⁺ cells are Caspase3⁺ in both control and *Top2b* mutant. After these observations, we focused on the analysis of electroporated granule cells in IGL. For this, we have extensively tested antibodies against genes that are supposed to be regulated by both Chd7 and Top2b according to our expression profiling data (see **Supplementary Table 3**). Due to the availability of specific antibodies for IHC, we could provide results from two Top2b target genes as below. Consistent with the fact that Top2b inhibition decreases the expression of Chd7 (**Figure 6d**), we observed significant decrease of Chd7 expression in Top2b sgRNA-expressing (GFP⁺) cells (**Figure 7b**). In addition, we showed a significant loss of expression of Zpfm2, a common target gene of Chd7 and Top2b, in GFP⁺ cells of sgTop2b cerebella (**Figure 7b**). These new data provide in vivo evidence demonstrating the requirement of Top2b for the expression of genes involved in neuronal differentiation of granule cells.

Due to the fact that only limited amount of progenitor cells (less than 10%) are targeted via initial electroporation, we do not expect to observe overall structure defects like hypoplasia of cerebellum in *Top2b* mutants.

REVIEWERS' COMMENTS:

Reviewer #2 (Remarks to the Author):

The authors have addressed my remaining concerns appropriately.

Reviewer #3 (Remarks to the Author):

The authors have addressed the most significant concerns raised, and have improved the paper substantially.